# Scale Size Estimation and Flow Pattern Recognition around a Magnetosheath Jet

Adrian Pöppelwerth[1], Georg Glebe[1,2], Johannes Z. D. Mieth[1], Florian Koller[3], Tomas Karlsson[4], Zoltán Vörös[5,6], and Ferdinand Plaschke[1]

[1]Institute of Geophysics and Extraterrestrial Physics, Technische Universität Braunschweig, Braunschweig, Germany
[2]School of Earth and Atmospheric Sciences, Georgia Institute of Technology, Atlanta, Georgia, USA
[3]Institute of Physics, University of Graz, Graz, Austria
[4]Division of Space and Plasma Physics, School of Electrical Engineering and Computer Science, KTH Royal Institute of Technology, Stockholm, Sweden
[5]Space Research Institute, Austrian Academy of Sciences, Graz, Austria
[6]Institute of Earth Physics and Space Science, HUN-REN, Sopron, Hungary

**Correspondence:** Adrian Pöppelwerth (a.poeppelwerth@tu-braunschweig.de)

**Abstract.** Transient enhancements in the dynamic pressure, so-called magnetosheath jets or simply jets, are abundantly found in the magnetosheath. They travel from the bow shock through the magnetosheath towards the magnetopause. On their way through the magnetosheath, jets disturb the ambient plasma. Multiple studies already investigated their scale size perpendicular to their propagation direction, and almost exclusively in a statistical manner. In this paper, we use multi-point measurements from the Time History of Events and Macroscale Interactions during Substorms (THEMIS) mission to study the passage of a single jet. The method described here allows us to estimate the spatial distribution of the dynamic pressure within the jet. Furthermore, the size perpendicular to the propagation direction can be estimated for different cross sections.

In the jet event investigated here, both the dynamic pressure and the perpendicular size increase along the propagation axis from the front part towards the center of the jet and decrease again towards the rear part but neither monotonically nor symmetrically. We obtain a maximum diameter in the perpendicular direction of about 1 $R_{\mathrm{E}}$ and a dynamic pressure of about 6 nPa at the jet center.

## 1 Introduction

The magnetic field of the Earth is an obstacle to the supersonic solar wind. To flow around the magnetopause, the boundary between the terrestrial and interplanetary magnetic fields (IMF), the solar wind must be decelerated to sub-magnetosonic speeds. This takes place upstream at the bow shock where the solar wind is decelerated, heated and deflected.

Depending on the angle $\theta_{\mathrm{Bn}}$ between the bow shock normal and the IMF, the bow shock can be divided into a quasi-parallel ($\theta_{\mathrm{Bn}} < 45°$) or quasi-perpendicular ($\theta_{\mathrm{Bn}} > 45°$) shock (e.g., Balogh et al., 2005). Particles reflected at the quasi-parallel shock can travel far upstream along the IMF and interact with the incoming solar wind. This leads to a region called foreshock which hosts a zoo of instabilities and waves (Eastwood et al., 2005). The waves are convected back to the shock with the solar wind, causing a rippled and undulated quasi-parallel bow shock.

The region between the bow shock and the magnetopause is called magnetosheath (e.g., Spreiter et al., 1966). In the magnetosheath, localized enhancements in the dynamic pressure are frequently observed. These so-called magnetosheath jets (see the review by Plaschke et al., 2018) were first reported by Němeček et al. (1998). Various definitions of jets can be found in the literature, which compare the dynamic pressure enhancement e.g. with the ambient plasma (e.g. Archer et al., 2012) or with the upstream solar wind (e.g. Plaschke et al., 2013). Jets are observed more often behind the quasi-parallel bow shock (e.g., Vuorinen et al., 2019) which corresponds to low IMF cone angle conditions for the subsolar magnetosheath and favor quiet solar wind (e.g., Plaschke et al., 2013). LaMoury et al. (2021) and Koller et al. (2023) further investigated the statistical dependence of jet occurrence on solar wind parameters. Jet impact rates determined by LaMoury et al. (2021) showed that more magnetosheath jets impact the magnetopause during low IMF magnitude, low solar wind density and high Mach number conditions. However, the dominant occurrence controlling parameters are low IMF cone angles and high solar wind speeds.

Jet formation downstream of the quasi-parallel bow shock may be explained by a mechanism suggested by Hietala et al. (2009, 2012). At the undulated bow shock, the incoming solar wind will be less decelerated and heated when passing the inclined parts. The geometry of the ripples can cause the flow to converge or diverge, resulting in density increases or decreases behind the shock. This leads to plasma regions with higher velocity and density than in the surrounding magnetosheath. Jets may also form due to solar wind discontinuities interacting with the bow shock (e.g., Archer et al., 2012). For example, Hot Flow Anomalies (HFAs, e.g. Savin et al., 2012) or short large amplitude magnetic structures (SLAMS, e.g. Schwartz and Burgess, 1991) can cause additional shock rippling when passing through the shock (e.g., Karlsson et al., 2018; Raptis et al., 2022b). That was also visible in simulations by Suni et al. (2021). They showed that jets can form due to the impact of compressional structures (like SLAMS) at the bow shock.

Geoeffective jets (with diameters $> 2R_{\mathrm{E}}$) reach the magnetopause several times per hour (Plaschke et al., 2020a) and have therefore a big impact on the magnetosphere and ionosphere. They can indent the magnetopause (e.g., Shue et al., 2009), cause surface waves (e.g., Archer et al., 2019) and may even penetrate through the boundary (e.g., Dmitriev and Suvorova, 2015). In addition, Němeček et al. (2023) showed in a statistical study that jets can be an explanation for extreme magnetopause positions and deviations from the model predictions. Hietala et al. (2018) showed that jets can trigger and suppress reconnection at the magnetopause, as they can modify the magnetic field in the magnetosheath and thus alter the shear angle at the magnetopause. This leads to situations where reconnection is triggered when it is not expected and vice versa (see also Vuorinen et al., 2021). Additionally, upon impact, jets can enhance ionospheric flow channels (Hietala et al., 2012) and disturb radio communication (Dmitriev and Suvorova, 2023). Nykyri et al. (2019) proposed that jets might even trigger substorms, leading to auroral brightenings. Also, Han et al. (2017) hypothesized in a statistical study that jets impacting the magnetopause are one possible source of throat auroras.

On their way from the bow shock to the magnetopause, plasma jets interact with the ambient magnetosheath plasma. Palmroth et al. (2021) used global hybrid-Vlasov simulations to study the evolution of jets inside the magnetosheath. They reported that the jets thermalize on their way to the magnetopause and become more 'magnetosheath-like' while they keep their propagation direction. In addition Raptis et al. (2022a) reported that jets may contain two plasma populations, a cold and fast jet and a hotter and slower background population. Not only the jets but also the ambient plasma is affected from the interaction.

Recent studies showed a slight alignment of the magnetic field along the jet propagation direction (Plaschke et al., 2020b) and a stirring of the magnetosheath plasma in the vicinity of the jet (Plaschke et al., 2017). Plaschke and Hietala (2018) reported in a statistical analysis that jets push slower plasma ahead of them and out of their way. Jets act like plows, and after their passage, the magnetosheath plasma fills the wake regions behind them. Plaschke and Hietala (2018) speculated that properties of jets like their scale size may influence the interaction.

Multiple studies report that magnetosheath jets have scale sizes in the order of 1 $R_{\mathrm{E}}$ in the directions parallel and perpendicular to the jet propagation. To obtain a simple estimation of the parallel size of a jet, it is sufficient to integrate the plasma velocity over the jet observation interval (Plaschke et al., 2020a) or multiply the duration of the jet interval with the maximum speed to get an upper size limit (Gunell et al., 2014). To obtain the perpendicular size at least two spacecraft are needed. Plaschke et al. (2016, 2020a) and Gunell et al. (2014) used pairs of spacecraft and derived the scale sizes in statistical studies from the probabilities for both spacecraft to observe a jet. Karlsson et al. (2012) used the four Cluster spacecraft (Escoubet et al., 2001) to investigate the scale sizes of single jets. The authors performed a minimum variance analysis to obtain a suitable, jet specific coordinate system. They extrapolated density profiles along these directions with linear fits allowing them to estimate the scale sizes in all 3 directions.

However, apart from Karlsson et al. (2012), all aforementioned authors used statistical analyses to obtain information on the scale sizes and other properties of magnetosheath jets. Here we show for the first time the spatial distribution of the dynamic pressure within different cross sections of a jet. To achieve this, we select a jet event observed by the Time History of Events and Macroscale Interactions during Substorms (THEMIS) spacecraft (Angelopoulos, 2008) and transform the velocity measurements in to a coordinate system aligned with the jet propagation direction. We use the vortical behavior of the plasma in the jet path (Plaschke and Hietala, 2018) to determine the position of the spacecraft within the plane perpendicular to the propagation direction. Ultimately, we use the positions and measurements of the spacecraft to estimate dynamic pressure profiles perpendicular to the propagation direction within different cross sections. In addition, we determine the perpendicular sizes with these profiles for different cross sections.

## 2   Data and Methods

We focus on a jet observed by the THEMIS A, D, and E spacecraft (THA, THD, THE) on 13 October 2010, around 16:04:00 UT. Measurements of the magnetic field (FGM, Auster et al., 2008), ion velocity, ion density, ion energy flux density and dynamic pressure (ESA, McFadden et al., 2008) in the GSE-X direction ($P_{\mathrm{dyn,x}}$) are shown in Fig. 1 in the rows from top to bottom (full moments in spin resolution). Following Plaschke et al. (2013), we label the point of maximum dynamic pressure ratio with reference to the upstream OMNI solar wind measurements (King and Papitashvili, 2005) as $t_{\mathrm{max}}$. Start and end times of the jet interval are labeled as $t_{\mathrm{start}}$ and $t_{\mathrm{end}}$, respectively. They denote the times where $P_{\mathrm{dyn,x}}$ equal one quarter of the solar wind dynamic pressure ($P_{\mathrm{dyn,sw}}$). The spacecraft THA, THD and THE observed the jet for 50 s, 66 s and 43 s, respectively.

The ion energy flux density (Fig. 1a4-c4) together with the high ion density (Fig. 1a3-c3) clearly show that all three spacecraft are in the magnetosheath at the time of the event. The positions in GSE coordinates are given above each column of the figure;

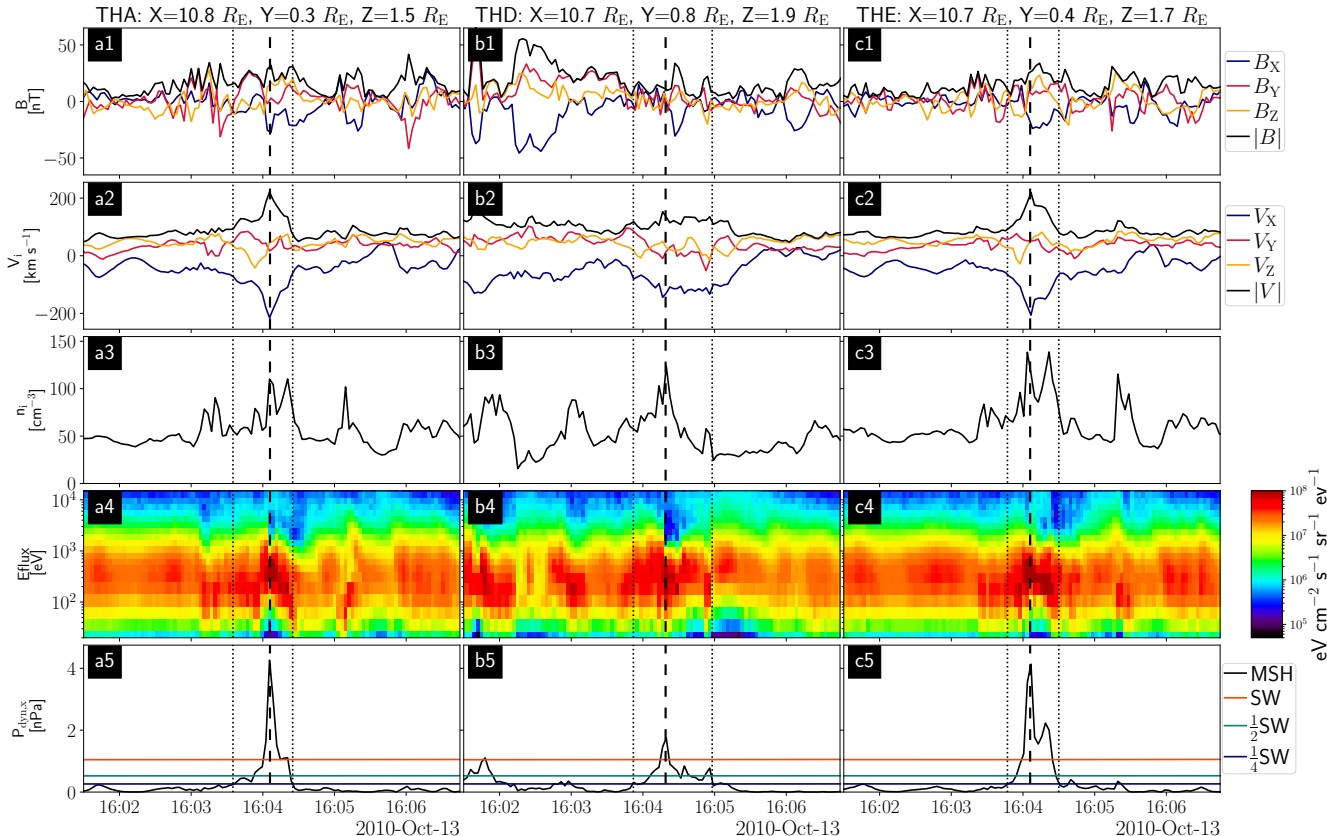

**Figure 1.** Plasma jet observed by the three THEMIS spacecraft THA (a), THD (b), and THE (c), respectively. From top to bottom, the magnetic field and ion velocity components in GSE coordinates and their magnitudes, the ion density, the ion energy flux density, and the GSE-X component of the dynamic pressure are shown. The vertical dashed lines in each column mark the times of maximum dynamic pressure ratio ($t_{\text{max}}$). The dotted lines denote the start ($t_{\text{start}}$) and end times ($t_{\text{end}}$) of the jet intervals. The horizontal lines in the last row represent the solar wind dynamic pressure, as well as half and a quarter thereof (in orange, cyan, and blue, respectively).

they show that all spacecraft are close to the sun-earth line. The dynamic pressure (Fig. 1a5-c5) exhibits a clear increase above the solar wind value for all spacecraft, ensuring that we are indeed observing a jet. The times $t_{\text{max}}$ are separated by only 13 s and the dynamic pressure peaks resulted from a combined increase in ion density and $V_{\text{x}}$ for every spacecraft. The increase in density is rather high compared to the statistics presented in Plaschke et al. (2013), as we observe an increase of about 100-200% from the ambient magnetosheath to the jet core around $t_{\text{max}}$. Although the velocities appear relatively low, they are still greater than half of the solar wind velocity while we are closer to the magnetopause.

A closer look at Fig. 1 shows a certain correlation of the individual components of the magnetic field $B$ and the ion velocity $V$ at THA and THE (Pearson correlation coefficients are between -0.34 and 0.92). Since the plasma beta is on the order of 10 within the jet and even higher outside, we can assume that the magnetic field lines align with the jet plasma flow, as discussed

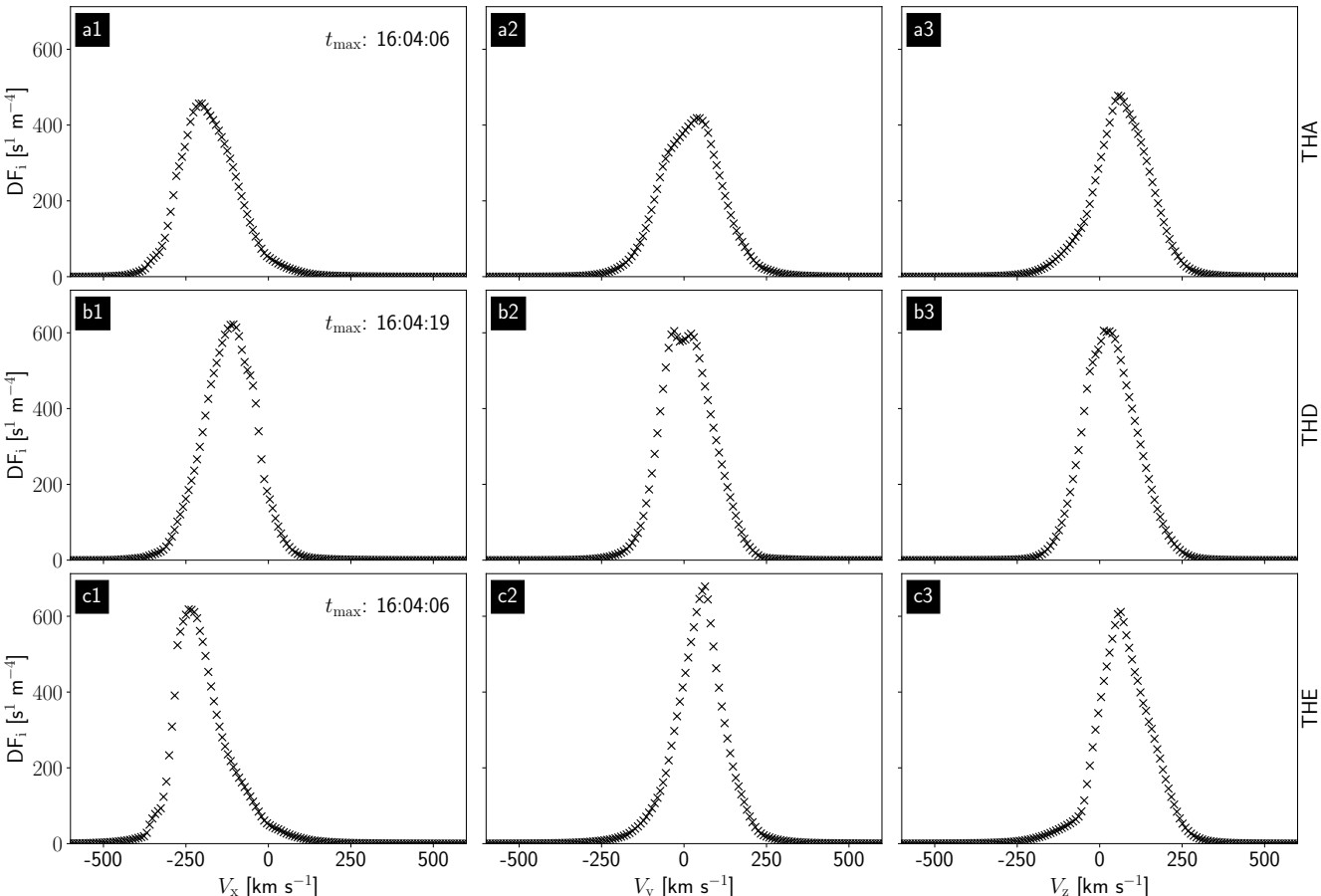

**Figure 2.** The integrated 1D velocity distribution function along the velocity components at the time of maximum dynamic pressure $t_{\mathrm{max}}$. The columns from left to right represent the $V_{\mathrm{x}}$, $V_{\mathrm{y}}$ and $V_{\mathrm{z}}$ component, respectively. The rows from top to bottom show the results for THA, THD and THE, respectively. In the left column we denote the time $t_{\mathrm{max}}$ for each spacecraft.

in Plaschke et al. (2020b). Outside the jet, in the ambient magnetosheath, the correlation is significantly lower (between -0.13 and 0.55).

Raptis et al. (2022a) have shown by investigating the velocity distribution function (VDF) that jets may contain a mixture of two plasma populations. In a similar manner we integrated the 3D VDF over two velocity axes to obtain a 1D VDF along the third velocity component. This is shown in Fig. 2 for the time $t_{\mathrm{max}}$ for all three velocity components of THA, THD and THE.

The rows from top to bottom show the VDFs for THA, THD and THE, respectively and the columns from left to right represent the $V_{\mathrm{x}}$, $V_{\mathrm{y}}$ and $V_{\mathrm{z}}$ component, respectively. We notice in agreement with Raptis et al. (2022a) that the $V_{\mathrm{y}}$ component

(VDF) of THD (see Fig. 2b2) shows two separate maxima. In addition the $V_{\mathrm{x}}$ and $V_{\mathrm{y}}$ components deviate from the ideal Maxwellian distribution. THA and THE measurements exhibit only single peaks in the 1D VDFs, but also deviate from the ideal Maxwellian distribution (see Fig. 2a1-a3 and Fig. 2c1-c3). The deviations of the 1D VDFs from the ideal Maxwell curve

could indicate that all three spacecraft are observing a mixture of jet and magnetosheath plasma. As THD is further away from the other two spacecraft and observes a lower dynamic pressure, we assume THD to be closer to the edge of the jet. It is even

possible that THD does not observe the jet, but the ambient magnetosheath, as the peak velocity in Fig. 2b1 is comparable to the background velocity in THA and THE (Fig. 2a1 and Fig. 2c1).

Therefore, we continue our analysis only for THA and THE by determining their jet velocities from the 1D VDFs similarly to Raptis et al. (2022a) in the following manner: We examine the 1D-VDFs at each time point and use the $V_x$, $V_y$ and $V_z$ values at which the 1D-VDFs have their maximum. If we observe deviations from the ideal Maxwellian (e.g. Fig. 2a1 or Fig. 2c1), we

choose the highest absolute velocity in the x-direction and the velocity of the coldest population for the y- and z-directions. We justify this procedure because the deviations from the ideal Maxwellian distribution indicate multiple plasma populations that may influence the moment calculations (for the influence of a background population on the moment calculation see: Raptis et al., 2022a).

To facilitate the analysis of the measurements, we need to define a coordinate system that is aligned with the direction of

120 the jet propagation. As the velocities are rather turbulent, we choose a short time (12 s) centered around $t_{max}$ and investigated the velocity directions (from the 1D VDFs) measured by THA and THE to determine the propagation direction. For an easier comparison of the directions we use spherical coordinates with the polar angle $\Theta$ and the azimuthal angle $\varphi$ to visualize the direction of the velocities:

$$\Theta = \arccos\left(\frac{z}{\sqrt{x^2 + y^2 + z^2}}\right), \quad \varphi = \mathrm{sgn}(y) \cdot \arccos\left(\frac{x}{\sqrt{x^2 + y^2}}\right). \tag{1}$$

The results are shown in Fig. 3 where the crosses in red, orange and blue represent the measurements of THA, THD and THE, respectively. THD is deviating strongly from THA and THE and is only shown for completeness. The black dot represents the mean value of the THA and THE measurements.

The directions of THA and THE are more similar and do not variate significantly around $t_{max}$. We therefore can assume that the mean values of $\Theta = 73.05°$ and $\varphi = 163.18°$ represent the propagation direction $V_{jet}$ well. In addition, we also calculate

the standard deviation ($\Theta = 4.04°, \varphi = 4.48°$) and maximum difference from the mean ($\Theta = 7.63°, \varphi = 9.00°$). To treat the uncertainty of the propagation direction conservatively, we will use the maximum differences as error estimation. With the mean values for $\Theta$ and $\varphi$ we determine the propagation direction $V_{jet} = [-0.92, 0.28, 0.29]$ (in GSE coordinates) and calculate the axes of the new coordinate system as follows:

$$X' = V_{jet}, \quad Y' = \frac{X' \times \hat{X}}{|X' \times \hat{X}|}, \quad Z' = \frac{X' \times Y'}{|X' \times Y'|}. \tag{2}$$

$\hat{X}$ is the unit vector along the GSE-X axis. $X'$ points in the propagation direction of the jet, while $Y'$ and $Z'$ are oriented perpendicular to the propagation direction and complete the right handed system. We choose the position of spacecraft THA at $t_{max}$ (see at the top of Fig. 1) as origin of our jet coordinate system since THA observes the highest dynamic pressure. To transform the velocities and positions we simply rotate them into the new coordinate system.

Using the jet coordinate system, we can investigate the flow patterns at the spacecraft positions. This is shown in Fig. 4,

where the arrows indicate the ion velocities (from the 1D VDFs) at the spacecraft positions (circles) of THA, THD and THE

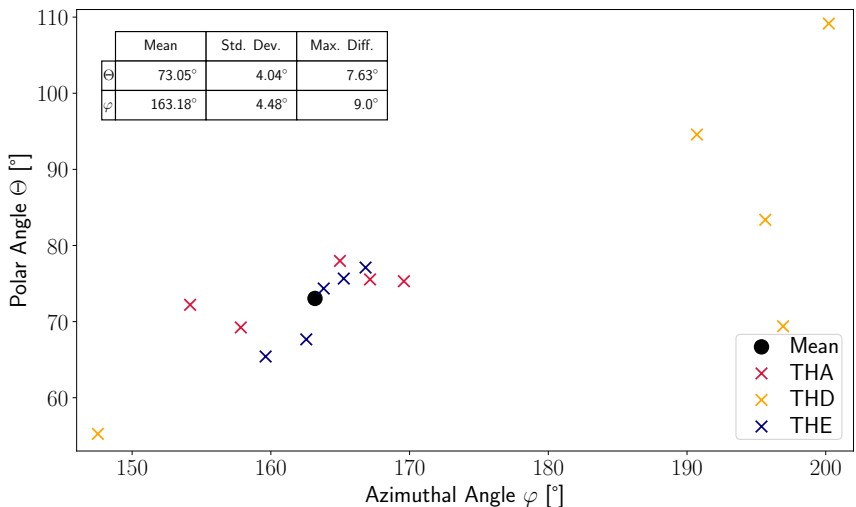

**Figure 3.** The polar angle $\Theta$ plotted against the azimuthal angle $\varphi$ for velocity measurements of THA, THD and THE around $t_{\max}$ in red, orange and blue, respectively. The black dot represents the mean value of the THA and THE measurements. The table in the upper right shows the mean values, standard deviations and maximum differences of $\Theta$ and $\varphi$.

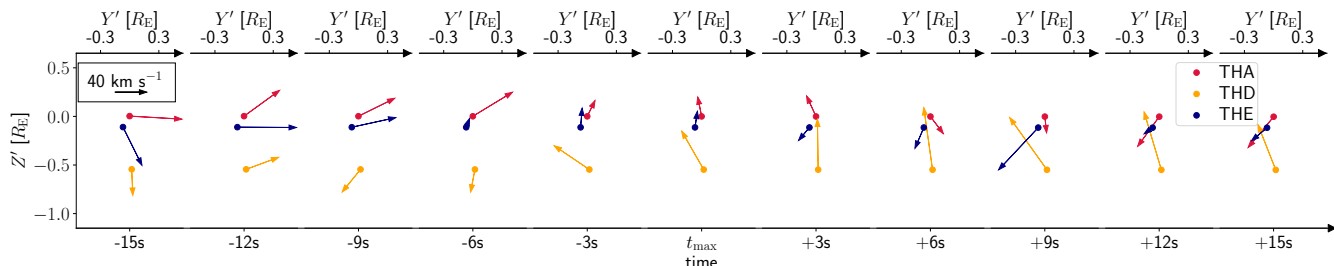

**Figure 4.** Ion velocities from the 1D VDFs at the three spacecraft positions for 11 time steps around $t_{\max}$ in the plane perpendicular to the jet propagation direction. The circles represent the spacecraft positions and the arrows indicate the velocities. The colors for THA, THD, and THE are red, orange, and blue, respectively. The top axis shows the corresponding $Y'$ coordinates for each time step, while the bottom axis displays the time steps. In the upper left corner, the black arrow indicates the scale.

in red, orange and blue, respectively. The figure shows the orientation of the velocities in the $Y' - Z'$ plane, perpendicular to the propagation direction, for 11 time steps from 15 s before to 15 s after $t_{\max}$. The $Y'$- and $Z'$-axis are identical for each time step.

Plaschke and Hietala (2018) have reported that the vortical motion of the plasma is not only visible outside of the jet but is also apparent within the jet structure. Therefore we choose this time range where all spacecraft observe the jet. We remind the reader that we will primarily focus on THA and THE as we have already discussed that THD is farther away from the other two spacecraft and might observes a mixture of plasma populations or even just the ambient magnetosheath. This said, we

argue that the following description applies also to THD but we expect deviations from the general behavior as the conditions are different compared to THA and THE.

On the left side of Fig. 4, prior to $t_{max}$, the arrows point towards the positive $\boldsymbol{Y'}$ direction, but in different $\boldsymbol{Z'}$ directions. We interpret this as signs of diverging flow. Closer to $t_{max}$, from 3 s before to 9 s after $t_{max}$, the arrows show a rather turbulent behavior and we observe rotations of the arrows, mostly in counterclockwise direction. Looking at Fig. 1a5 and c5 we see two high dynamic pressure peaks in this time interval at THA and THE. In contrast, the arrows on the right side in Fig. 4, from -12 s to -15 s, corresponding with times after $t_{max}$, point towards the negative $\boldsymbol{Y'}$ direction. The arrows point additionally to roughly one point and show signs of a converging plasma flows.

In Appendix A we show that the visibility of this vortical motion is not strongly dependent on the propagation direction, as the same flow patterns are still visible at THA and THE for slightly rotated coordinate systems that are consistent with the determined uncertainties.

Next we determine which regions of the jet the spacecraft observe. In order to achieve this, we use the diverging flows before and the converging flows after $t_{max}$ to estimate the position of the central axis of the jet. Thereafter, we can calculate the spacecraft distances from the central axis within different cross sections of the jet.

We extend the THA- and THE- velocity vectors in the $\boldsymbol{Y'}$-$\boldsymbol{Z'}$-plane and determine the central axis as the point where the two lines intersect. As an example, in Fig. 5a, b we show the estimation for the time steps 12 s before and 15 s after $t_{max}$. The gray lines indicate the extension of the velocity vectors and the black cross represents the estimated position of the central axis. In Fig. 5c we present the estimated positions of the central axis for all time steps, except where we observe the largest dynamic pressures (from 3 s before to 9 s after $t_{max}$).

Based on Fig. 5c, we can calculate the mean position of the central axis: $Y' = -0.15\ R_E$ and $Z' = -0.10\ R_E$. We also observe that the position is relatively well determined, as can be seen by the low standard deviations ($Y' = 0.10\ R_E$, $Z' = 0.07$ $R_E$) and maximum differences ($Y' = 0.20\ R_E$, $Z' = 0.11\ R_E$). Only the estimation at 9 s prior to $t_{max}$ causes a large error, especially in the $Y'$ direction. Again, to be conservative, we use the maximum differences as uncertainties and assume the position of the central axis to be valid for the entire jet interval.

In the next section we will use the spacecraft positions and their distances from the central axis to investigate the dynamic pressure profiles for different cross sections. In order to achieve this, we fit a Gaussian distribution that also can be used to estimate the scale size of the corresponding cross section to the $P_{dyn}$ measurements:

$$P_{dyn,fit} = P_0 \cdot \exp\left(\frac{-r^2}{2 \cdot \Delta R^2}\right). \tag{3}$$

Here the parameters $P_0$ and $\Delta R$ are the amplitude and width of the Gaussian, and $r$ represents the distance to the central axis. The choice of the Gaussian profile may be somewhat arbitrary. Even though we cannot guarantee that it describes the jets in reality, the measurements in this case are well described by this profile. To apply this fit we have to assume a monotonous decrease of the dynamic pressure from the center towards the edges in the direction perpendicular to the jet propagation (with increasing $r$). We furthermore assume a rotational symmetry around the central jet axis to ensure a robust fit. However, we do not make any assumptions for the dynamic pressure along the propagation axis.

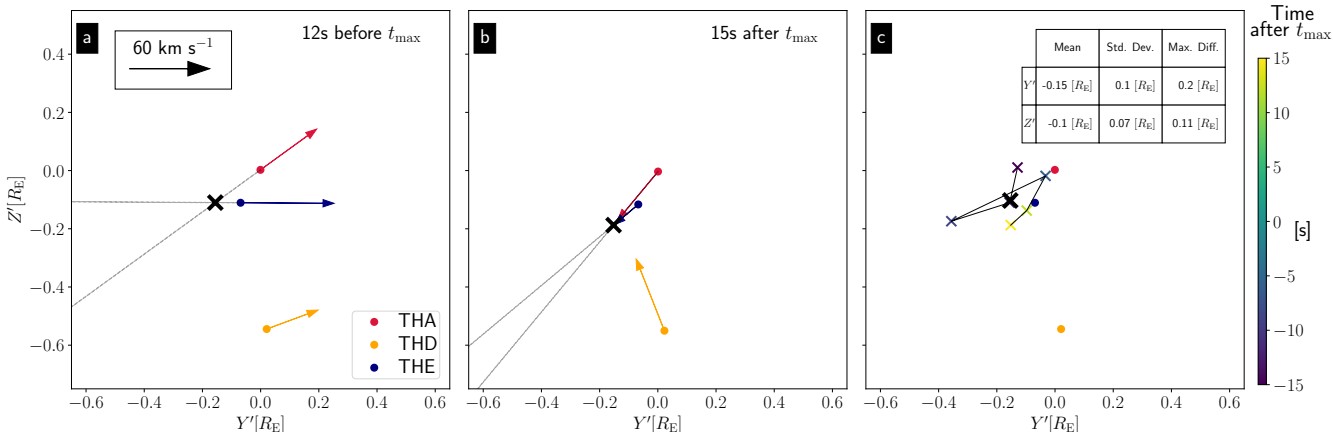

**Figure 5.** a) and b): Ion velocities from 1D VDF peaks at the three spacecraft positions 12 s before (a) and 15 s after $t_{\mathrm{max}}$ (b) in the plane perpendicular to the jet propagation. The circles represent the positions of the spacecraft and the arrows indicate the velocities. The colors for THA, THD, and THE are red, orange, and blue, respectively. The black arrow indicates the scale. The gray lines are simple extensions of the velocity vectors, and the black crosses mark the intersections of the lines, and representing the estimated center positions. c): The colored crosses show the estimated positions for different time steps. The color corresponds to the time as indicated by the color bar. The black cross represents the mean value and the dots in red, orange and blue are the positions of THA, THD and THE, respectively. The table in the upper right denotes the mean values, standard deviations and maximum differences of the $Y'$ and $Z'$ coordinates.

## 3  Results and Discussion

Using the estimated position of the central axis, we calculate the distances of the spacecraft from the central axis $r$ in the $Y'$-$Z'$-plane. This results in distances of 0.19 $R_{\mathrm{E}}$, 0.48 $R_{\mathrm{E}}$ and 0.09 $R_{\mathrm{E}}$ for THA, THD and THE, respectively. These values
change only marginally (max. 3%) over the jet interval due to the spacecraft movement, assuming the central axis staying constant.

To obtain dynamic pressure profiles we, plot $P_{\mathrm{dyn,x}}$ derived from the velocities from the 1D VDFs in the spacecraft system against the distances $r$ for different times and apply the Gaussian fit. In Fig.6 we show this for the times $t_{\mathrm{max}}-9$ s (a), $t_{\mathrm{max}}$ (b) and $t_{\mathrm{max}}+15$ s (c). Crosses in red, orange, and blue represent the data points for THA, THD and THE, respectively. We also
plot one quarter of the solar wind dynamic pressure (blue horizontal line) in Fig.6a-c and the Gaussian distribution is shown as black, dashed line. The gray area visualizes the standard deviation $\sigma$ of the optimal fit parameters. Here, $\sigma$ is the square root of the diagonal elements of the covariance matrix for the fitting parameters.

In the three time steps shown, the dynamic pressure is highest at the spacecraft closest to the center (THE). While we see some deviations from the data in Fig.6b (larger gray area), the fit in Fig.6a and Fig.6c represents the data points very well. The
fit parameters are $P_0 =0.79 \pm 0.02$ nPa and $\Delta R =0.27 \pm 0.01$ $R_{\mathrm{E}}$ at $t_{\mathrm{max}}-9$s, $P_0 =6.33 \pm 0.60$ nPa and $\Delta R =0.27 \pm 0.04$ $R_{\mathrm{E}}$ at $t_{\mathrm{max}}$ and $P_0 =2.39 \pm 0.32$ nPa and $\Delta R =0.15 \pm 0.02$ $R_{\mathrm{E}}$ at $t_{\mathrm{max}}+15$s. The estimated central jet dynamic pressure is higher at $t_{\mathrm{max}}$ (6 nPa) than earlier at $t_{\mathrm{max}}-9$s (1 nPa) or later at $t_{\mathrm{max}}+15$s (2 nPa).

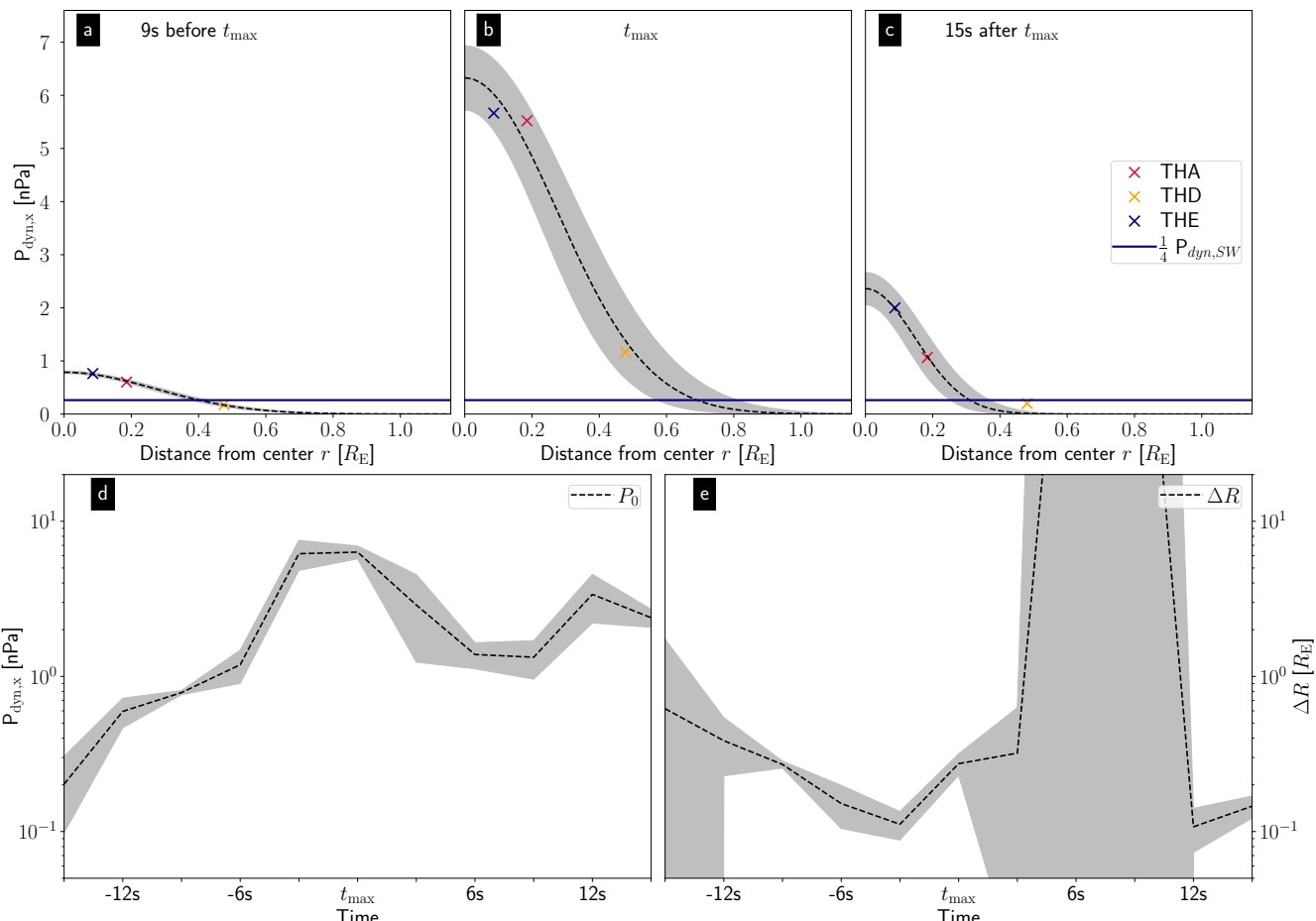

**Figure 6.** Dynamic pressure $P_{\mathrm{dyn,x}}$ derived from velocities from the 1D VDFs in the spacecraft system versus the distance from the center $r$ at THA, THD and THE (crosses in red, orange and blue, respectively) at 9 s before $t_{\max}$ (a), at $t_{\max}$ (b) and at 15 s after $t_{\max}$ (c). The black, dashed line represent a fit with a Gaussian distribution to the data points. The blue horizontal line depicts a quarter of the solar wind dynamic pressure. In d) and e) we display the development of the fit parameter $P_0$ and $\Delta R$, respectively. The gray area visualizes in all panels when we subtract/add one standard deviation $\sigma$ from the optimal fit parameters.

Furthermore, we show the evolution of the central axis dynamic pressure $P_0$ and the width of the Gaussian fit $\Delta R$, obtained by fitting the Eq. 3 to the data for the different time steps, in Fig. 6d and Fig. 6e, respectively. In both panels the optimal fit parameter are shown in black and the gray areas visualize the standard deviation $\sigma$ of the optimal fit parameters. We observe an increase of the dynamic pressure $P_0$ from $t_{\max}$-15 s to $t_{\max}$ followed by an decrease thereafter. In addition, we can recognize a second peak at $t_{\max}$+12 s, which is already visible in Fig. 1c5 and partially in Fig. 1a5. The increase and decrease of the dynamic pressure along the central jet axis are neither symmetric nor monotonic. The width of the Gaussian profile $\Delta R$ shows no clear trend within the jet, but two extreme outliers where the fit was not appropriate (6 s and 9 s after $t_{\max}$). At these times,

THD observed higher $P_{\mathrm{dyn,x}}$ values than THA, despite being farther away from the central axis, resulting in an unrealistic width of the Gaussian fit. On the left-hand side in Fig. 6e, $\Delta R$ decreases until 3 s before $t_{\max}$. It then increases even beyond $t_{\max}$ and is quite low at the end after the two outliers. The uncertainty of $\Delta R$ is greater compared to $P_0$, which is indicated by the larger gray area. In contrast, the values of $\Delta R$ only vary by about one order of magnitude (excluding the two outliers), while the values of $P_0$ change more strongly between approximately two orders of magnitude.

The intersection of the fit with $P_{\mathrm{dyn,x}} = \frac{1}{4} P_{\mathrm{dyn,sw}} = 0.26$ nPa leads to an estimation of the jet size in the direction perpendicular to the jet propagation. We choose one quarter of the solar wind dynamic pressure as a threshold to be consistent with the definition of $t_{\mathrm{start}}$ and $t_{\mathrm{end}}$ for jets, which determine the scale size along the jet propagation direction (see criterion of Plaschke et al., 2013). The Gaussian fits (black lines) intersect the horizontal line at 0.40 $R_{\mathrm{E}}$, 0.69 $R_{\mathrm{E}}$ and 0.31 $R_{\mathrm{E}}$ at $t_{\max}-9$ s, $t_{\max}$ and $t_{\max}+15$ s, respectively. It is essential to note that both $P_0$ and $\Delta R$ contribute to the perpendicular size and not
one of them alone can describe it. The shape is therefore quite complex as we observe contrary increases and decreases of $P_0$ and $\Delta R$.

For instance, the width of the Gaussian distribution $\Delta R$ is quite similar in the front and central part, but the higher dynamic pressure in the jet center results in a larger perpendicular size of the jet around $t_{\max}$. Similarly, the lower dynamic pressure at the front part results in a smaller perpendicular extension. On the other hand, the comparison between the front part ($t_{\max}-9$
220 s) and the rear part ($t_{\max}+15$ s) shows the opposite behavior. We observe a larger perpendicular extension in the front part, although the central dynamic pressure is higher in the rear part. In this case, the greater width of the Gaussian distribution $\Delta R$ at $t_{\max}-9$ s leads to the larger perpendicular size.

We applied the fit here to the measurements from all three spacecraft to reduce the uncertainty of the parameters and provide an estimate of the uncertainty, which is not possible when fitting to only two data points. However, the qualitative results
remain the same if we use only the dynamic pressure measurements from THA and THE. Thus, the possibility that THD is not observing the plasma of the jet but the ambient magnetosheath does not change the conclusions we can draw from our method applied to this jet. In Appendix B we show that the dynamic pressure profiles do not strongly depend on the position of the central axis, as the parameters $P_0$ and $\Delta R$ and their evolution over the jet interval vary only marginally with varying central axis position.

We can compare the estimated scale sizes with previous results. In previous studies, different authors reported a range of scale sizes of magnetosheath jets. Plaschke et al. (2020a) derived that most of the jets should be on the order of 0.1 $R_{\mathrm{E}}$, although they argued that these small jets are less likely to be observed. For the observed magnetosheath jets, they report a median diameter of about 1 $R_{\mathrm{E}}$ in the directions parallel and perpendicular to the flow. Gunell et al. (2014) calculated upper limits and found median values of 4.9 $R_{\mathrm{E}}$ and 3.6 $R_{\mathrm{E}}$ for the sizes parallel and perpendicular to the flow, respectively. Both
studies used pairs of spacecraft and the probabilities that both observe a jet to calculate sizes perpendicular to the propagation directions. Karlsson et al. (2012) found scale sizes between 0.1 and 10 $R_{\mathrm{E}}$ for one direction perpendicular to the magnetic field; for the other two dimensions the sizes were found to be a factor of 3-10 larger. Thus our results with diameters of approximately 1.3 $R_{\mathrm{E}}$ at $t_{\max}$ and 0.8 $R_{\mathrm{E}}$ at times before and after $t_{\max}$ fit very well to the earlier reported sizes.

The method presented by Karlsson et al. (2012) can be used to obtain the sizes of single jets in all three dimensions. This is only possible if the structure is associated with a magnetic field discontinuity, which was the case for all their events. Contrary to this, our method provides scale sizes for the directions parallel and perpendicular to the flow under the assumption of rotational symmetry and a constant propagation direction. We have shown that the latter is given for this jet event to some extent. Furthermore, we assume radial dynamic pressure profiles that resemble Gaussian distributions. The problem can thus be reduced to two dimensions. This enables us estimate the perpendicular scale size for different cross sections of a jet. Together with the parallel scale size, we could create a simple 3D model of the magnetosheath jet. To apply this method, it is necessary to observe the flow pattern described by Plaschke and Hietala (2018). At least one of the two motions - diverging or converging plasma flow - should be visible to determine the position of the jet central axis. Observing both parts of the vortical motion leads to more reliable results. This estimation is therefore not applicable to all jets observed by multiple spacecraft, as individual events can deviate strongly from the average behavior. As Plaschke et al. (2020b) have shown for the alignment of velocity and magnetic field, the fluctuations can easily be on the same order of magnitude as the average alignment effect.

Note that the method described in this paper relies on above mentioned assumptions and simplifications. The choice of the Gaussian distribution for the fit implies a corresponding monotonous decrease of the dynamic pressure from the center towards the edges. These assumptions may not necessarily be satisfied in general or in some parts of the jet. To perform our analysis we need at least two spacecraft. However, as evident for this jet event, this is not necessarily sufficient, as the spacecraft should be well separated from each other. More spacecraft observing the same jet or the ambient magnetosheath would allow an evaluation of the validity of our assumptions.

## 4 Summary and Conclusion

In this paper we demonstrate a new method to determine the size for single jet events using in principle measurements from only two spacecraft. Here we have observed the vortical motion of plasma within a jet with the three THEMIS spacecraft THA, THD and THE. From the diverging flows ahead of and the converging plasma flows behind the jets maximum dynamic pressure region (at $t_{\mathrm{max}}$) we were able to estimate the position of the jet central axis. The distances of the spacecraft from the central axis were used together with the measured $P_{\mathrm{dyn,x}}$ to fit Gaussian distributions. This allowed us to determine the dynamic pressure profiles and the perpendicular sizes of the jet within different cross sections.

We have presented here dynamic pressure profiles for the jet event for three different times ($t_{\mathrm{max}}$-9 s, $t_{\mathrm{max}}$ and $t_{\mathrm{max}}$+15 s). Together with the development of the fit parameters $P_0$ and $\Delta R$ we can draw the following conclusions for this event:

1. The dynamic pressure in the central part of the jet is higher at $t_{\mathrm{max}}$ (6 nPa) and decreases towards the front and rear parts. But the increase and decrease are neither monotonic nor symmetrical. In addition, we observed a second peak after $t_{\mathrm{max}}$.

2. The width of the Gaussian distribution and the central dynamic pressure are variable over the jet interval. This results in a rather complex shape with varying diameter along the propagation axis. We observed the largest perpendicular size at $t_{\mathrm{max}}$ (1.2 $R_{\mathrm{E}}$) due to the high dynamic pressure in the center.

In this paper we cannot explain the asymmetric and non-monotonic increase and decrease of the dynamic pressure along the central axis and also not the variations in the width of the Gaussian profile. Future work could therefore focus on the evolution of jets and small-scale structures within the jet. However, it may be advantageous or necessary to use spacecraft data with a higher resolution for this task.

The apparent larger scale size around $t_{\max}$ suggests that some spacecraft may only observe central parts of a jet rather than the front and rear parts when passing through edge regions. In addition, spacecraft will unlikely observe the exact center of a jet. Thus, they would measure just a fraction of the dynamic pressure in the jet center (a lower limit) as $P_{\mathrm{dyn,x}}$ decreases towards the edges and this would not necessarily be representative for the jet. This implies that statistical studies of dynamic pressures of jets may significantly and systematically underestimate the maximum values (e.g., Raptis et al., 2020). Furthermore, we also emphasize that the comparison of observations with simulations for jets and in general transient and localized phenomena in any plasma environment must take this bias into account.

The jet event selected for this case study belongs to a fraction of jet observations that show clear signs of the expected flow pattern that is needed for the estimation of the central axis. Furthermore, this jet event appears to be quite rare, as we observe very high densities and comparatively low velocities. Other events may differ quantitatively from this case, but we see no reason why it should not work for them. We would like to remind the reader once again that this case study only demonstrates the concept of determining the jet size of individual events and does not claim to determine the general shape of jets.

With only three spacecraft available, there are uncertainties regarding the quality and applicability of the fit and validity of our assumption of rotational symmetry. To increase our confidence in the fit and our assumptions, it would be useful to obtain measurements from even more spacecraft on a jet. This could be achieved through conjunctions of spacecraft from different missions like Cluster (Escoubet et al., 2001), Magnetospheric Multiscale (MMS, Burch et al., 2016) and THEMIS (Angelopoulos, 2008).

## Appendix A: Uncertainty of $V_{\mathrm{jet}}$

The propagation direction of the jet ($V_{\mathrm{jet}}$) may have a great impact on our analysis. If the direction is incorrect or poorly determined, it is possible that we do not notice the vortical movement even though it is actually present or vice versa. We estimated $V_{\mathrm{jet}}$ as a mean value of multiple measurements by THA and THE. Thus, we imply a constant propagation direction over time and that the velocities at both spacecraft positions represent the propagation well. To handle the uncertainty of this assumptions, we take a look at the maximum differences from the mean velocity (cf. Fig. 3). Adding or subtracting these values to/from $V_{\mathrm{jet}}$ result in $V_{\mathrm{jet,max}}$ and $V_{\mathrm{jet,min}}$ as alternative propagation directions.

With $V_{\mathrm{jet,max}}$ and $V_{\mathrm{jet,min}}$ we can transform the measured ion velocities and the positions of the spacecraft into new coordinate systems using Eq. 2. We than look at the transformed velocities in the $Y' - Z'$ plane (perpendicular to the propagation direction). This is shown in Fig. A1 for $V_{\mathrm{jet,max}}$ in the top row, for $V_{\mathrm{jet,min}}$ in the bottom row and for $V_{\mathrm{jet}}$ in the middle row (for comparison).

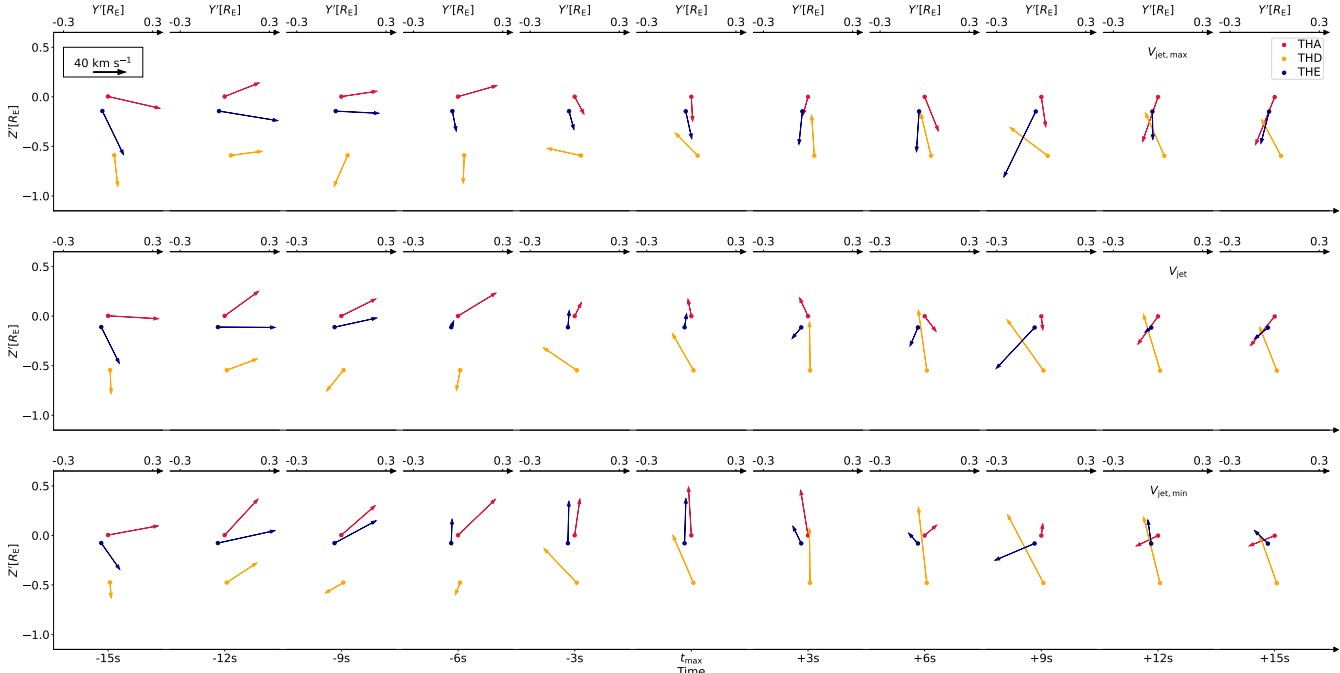

**Figure A1.** Ion velocities from the 1D VDFs at the three spacecraft positions for 11 time steps around $t_{\mathrm{max}}$ in the plane perpendicular to the jet propagation direction. The circles represent the spacecraft positions and the arrows indicate the velocities. The colors for THA, THD, and THE are red, orange, and blue, respectively. The top axes show the corresponding $\boldsymbol{Y'}$ coordinates for each time step, while the bottom axes display the time steps. In the upper left corner, the black arrow indicates the scale. The top, middle and bottom row were calculated with $\boldsymbol{V_{\mathrm{jet,max}}}$, $\boldsymbol{V_{\mathrm{jet}}}$ and $\boldsymbol{V_{\mathrm{jet,min}}}$ as propagation direction, respectively.

We observe the diverging flows before and the converging flows after $t_{\mathrm{max}}$ in all three cases. In addition, we also investigate
if the use of the velocities from the 1D VDFs has an influence on our results. Therefore we use the same propagation directions $\boldsymbol{V_{\mathrm{jet,max}}}$, $\boldsymbol{V_{\mathrm{jet}}}$ and $\boldsymbol{V_{\mathrm{jet,min}}}$ and transform the ion velocities calculated from the full moments. This is shown in Fig. A2.

Although there are some changes, we can again observe in all cases the diverging flows before and the converging flows after $t_{\mathrm{max}}$. Thus we conclude that the flow pattern we observe is not an artifact from our data handling.

### Appendix B: Uncertainty of jet center estimation

Since THA and THE are close to each other and in the vicinity of the central axis in the plane perpendicular to the propagation direction, minor deviations of the position of this axis can have major effects on the dynamic pressure profiles. Therefore, we use the maximum differences from the mean to calculate alternative positions of the central axis and compare the resulting pressure profiles. In order to achieve this, we look at the fit parameters $P_0$ and $\Delta R$ and how these values change with varying central axis positions. In addition, we investigate again whether the use of the velocities from the 1D VDFs has a major

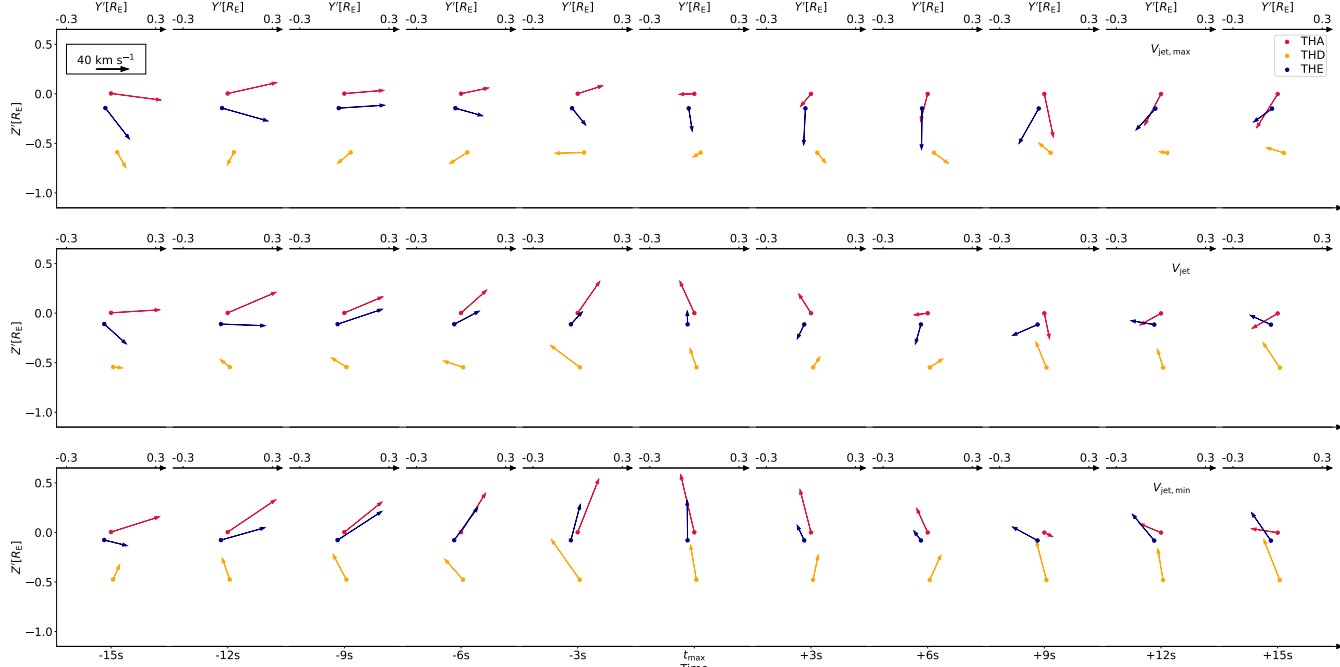

**Figure A2.** Ion velocities from the full moments at the three spacecraft positions for 11 time steps around $t_{\mathrm{max}}$ in the plane perpendicular to the jet propagation direction. The circles represent the spacecraft positions and the arrows indicate the velocities. The colors for THA, THD, and THE are red, orange, and blue, respectively. The top axes show the corresponding $\boldsymbol{Y'}$ coordinates for each time step, while the bottom axes display the time steps. In the upper left corner, the black arrow indicates the scale. The top, middle and bottom row were calculated with $\boldsymbol{V_{\mathrm{jet,max}}}$, $\boldsymbol{V_{\mathrm{jet}}}$ and $\boldsymbol{V_{\mathrm{jet,min}}}$ as propagation direction, respectively.

influence on the results. Therefore, we calculate $P_{\mathrm{dyn,x}}$ from the velocities from the full moments and repeat the comparison. These results are presented in Fig. B1.

The panels (from top to bottom) show the time evolution of the fit parameters $P_0$ (a1,a2) and $\Delta R$ (b1,b2) and the time evolution of their uncertainties $\sigma(P_0)$ (c1,c2) and $\sigma(\Delta R)$ (d1,d2). The left and right column show parameters for the dynamic pressure calculated with the velocities from the 1D VDFs and with the velocities from the full moments, respectively. The lines

represent the results with the mean central axis (solid) and the mean central axis with errors subtracted/added (dotted/dashed).

The dynamic pressure at the central axis is highest around $t_{\mathrm{max}}$ for all cases. We do not observe the second peak at $t_{\mathrm{max}}$+12 s if we add the errors. Looking at Fig. B1c, we can see that the uncertainty of the fit parameter $\sigma(P_0)$ is peaks at $t_{\mathrm{max}}$-3 s and $t_{\mathrm{max}}$+12 s. The latter may explain why we do not observe the second peak in $P_0$.

For $\Delta R$ (Fig. B1c) we observe again a rather similar trend for all cases over the whole time with some exceptions that

correlate well with higher uncertainties in $\sigma(\Delta R)$ (Fig. B1d). These extremely high uncertainties arise when the data do not show a monotonic decrease in dynamic pressure and a Gaussian fit is not appropriate. Therefore, we argue that the exact position of the central axis does not have a major impact on our conclusion.

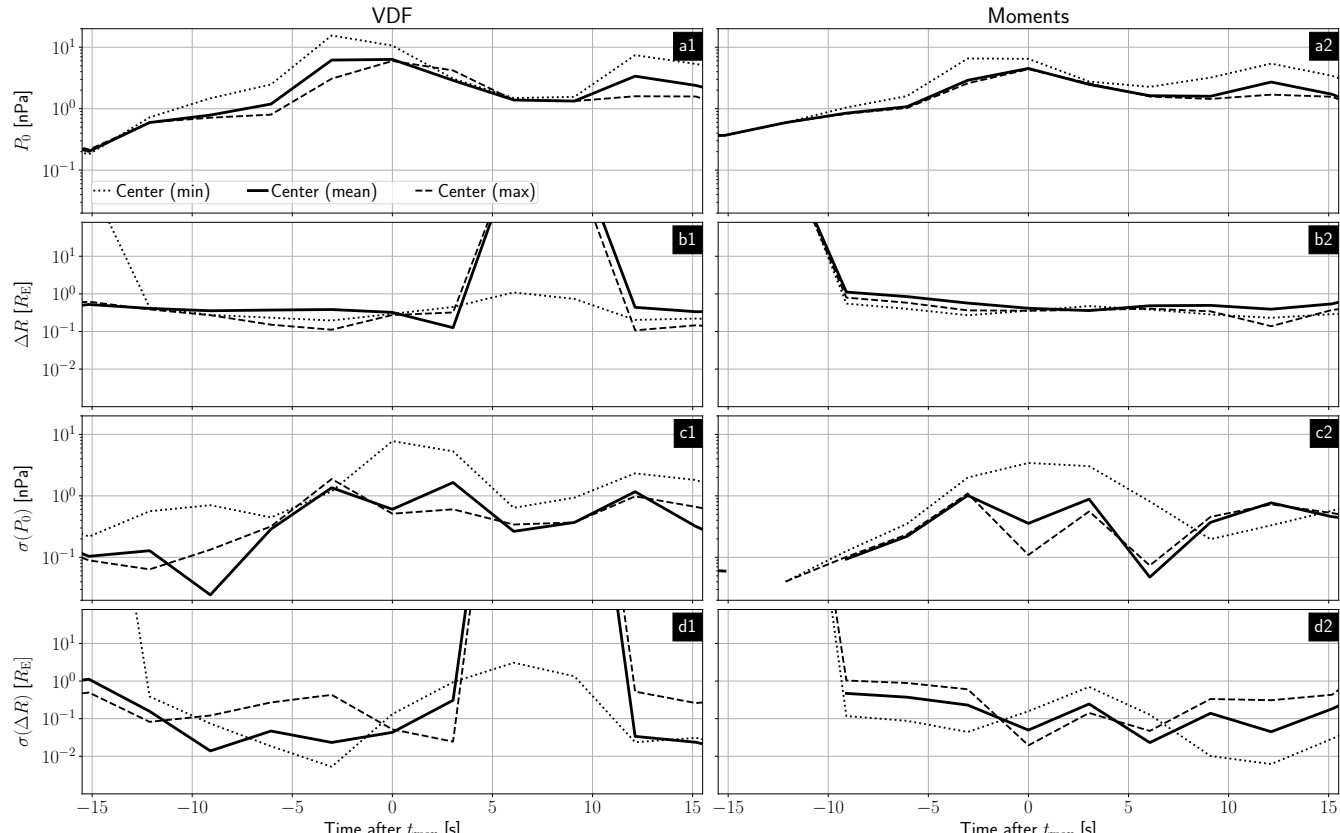

**Figure B1.** The lines represent the results with the mean central axis (solid), mean central axis with errors subtracted (dotted) and errors added (dashed). The left and right column show parameters for the dynamic pressure calculated with the velocities from the 1D VDFs and with the velocities from the full moments, respectively. The panels from top to bottom show the time evolution of the fit parameters $P_0$ (a1,a2) and $\Delta R$ (b1,b2) and the time evolution of their uncertainties $\sigma(P_0)$ (c1,c2) and $\sigma(\Delta R)$ (d1,d2).

*Data availability.* Data from the THEMIS mission including level 2 FGM and ESA data are publicly available from the University of California Berkeley and can be obtained from http://themis.ssl.berkeley.edu/data/themis (THEMIS, 2024). The solar wind data from NASA's OMNI high-resolution data set (1 min cadence) are also publicly available and can be obtained from https://spdf.gsfc.nasa.gov/pub/data/omni/omni_cdaweb (OMNI, 2024). THEMIS and OMNI data were accessed using the PySPEDAS software (Grimes et al., 2018; Angelopoulos et al., 2019).

*Author contributions.* AP performed the main work. GG, FK, TK, ZV and FP helped with the discussions and interpretations of the results. JM took care of THEMIS FGM calibrations and brought his expertise on THEMIS data into the discussions.

*Competing interests.* The authors declare that they have no conflict of interest.

*Acknowledgements.* We acknowledge NASA contract NAS5-02099 for use of data from the THEMIS Mission, specifically C.W. Carlson and J. P. McFadden for the use of ESA data; K. H. Glassmeier, U. Auster, and W. Baumjohann for the use of FGM data provided under the lead of the Technical University of Braunschweig and with financial support through the German Ministry for Economy and Technology and the German Center for Aviation and Space (DLR) under contract 50 OC 0302. This work was financially supported by the German Center
for Aviation and Space (DLR) under contract 50 OC 2201. FK and ZV acknowledge the support by the Austrian Science Fund (FWF), P 33285-N.

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
