# Peer review of "Scale Size Estimation and Flow Pattern Recognition around a Magnetosheath Jet"

_Annales Geophysicae, 2023_

## Author Response (AR1)

In this document we give responses to comments from both referees. We start with referee#1 and then answer referee #2. At the end we provide a list with relevant changes made in the manuscript.

**Author response to referee #1**

The authors thank Anonymous referee #1 for their comments on the manuscript. We will take the comments into account when revising the manuscript. Here we provide responses to each of the referee's comments (formatted as italics in indented paragraphs). At the end of our answers, we indicate in blue where the changes have been made in the revised version.

> *The weakness of the method is the assumption that the direction of the jet velocity does not change in course of its observation and it can lead to misinterpretation of measurements. Moreover, the data processing does not handle with uncertainties of velocity measurements that are rather large because the velocity distribution in the magnetosheath is hardly Maxwellian, especially when the jets are present. These factors result to uncertainty in a determination of the jet coordinate system and it is fully possible that the lack of divergence in front of jet discussed in the first paragraph of Discussion section is caused by these factors. A small rotation of the coordinate system probably would lead to the diverging patterns in front of the jet but it is a question whether the converging patterns at its end will persist under assumption of the constant jet velocity.*

We agree that the treatment of uncertainties is missing and that the full moments are error-prone for our analysis. Therefore, we repeated our analysis considering the velocity distribution functions (VDFs) and included error handling. We started with a closer look at the velocity distribution functions. A first glance, the 2D slices in the plane $V_X - V_Y$ shows only single maxima for THA, THD and THE, as shown in Figure 1.

[Figure]

| (a) THA | (b) THD | (c) THE |

Figure 1: 2D slices of the velocity distribution around $t_{max}$ in the $V_X - V_Y$ plane at $V_Z = 0$.

For a more detailed look, we used a similar approach as Raptis et al. [2022] and integrated the 3D distribution over two velocity axes to obtain a 1D velocity distribution along the third component. The 1D distributions along $V_X$, $V_Y$ and $V_Z$ (in GSE coordinates) at the time of maximum dynamic pressure ($t_{max}$) are shown in Figures 2, 3 and 4 for THA, THD and THE, respectively.

Here, we were not able to see a clear bi-Maxwellian distribution for THA and THE in either of the three components [cf. Raptis et al., 2022]. Instead, we observe individual peaks, albeit with deviations from the ideal Maxwellian distribution. For THD, we clearly see two maxima in the 1D distribution along $V_Y$.

Following the same conclusion as Raptis et al. [2022], THA and THE likely observe one plasma population. THD, which is also further away from the other two, observes two plasma populations, probably a mixture of jet and magnetosheath background plasma. Together with the lower dynamic pressure at $t_{max}$, this suggests that THD is closer to the edge of the jet. To be conservative, we continued our analysis using only THA and THE to determine the direction of jet propagation. Furthermore, since we see that the 1D VDFs at THA and THE deviate from ideal Maxwellians, we use the velocity at which the 1D VDF peaks in each component rather than the full plasma moments to determine the jet propagation direction. Other time steps inside the jet show similar 1D VDFs and outside the jet we observe 1D VDFs that look almost like ideal Maxwellians.

[Figure]

(a) THA - $V_X$            (b) THA - $V_Y$            (c) THA - $V_Z$

Figure 2: 1D distribution from integration along the other velocity axes.

[Figure]

(a) THD - $V_X$            (b) THD - $V_Y$            (c) THD - $V_Z$

Figure 3: 1D distribution from integration along the other velocity axes.

As the referee correctly stated, the direction of the jet velocity can change during the course of the observation. We therefore investigated the stability of the direction of the jet velocity. Using the updated velocity values, we calculated the polar angle $\Theta$ $(= \arccos \frac{V_z}{\sqrt{V_x^2+V_y^2+V_z^2}})$ and the azimuthal angle $\varphi$ $(= \mathrm{sgn}(V_y) \cdot \arccos \frac{V_x}{\sqrt{V_x^2+V_y^2}})$ of the measured velocities and plotted them over time. The results are shown in Figure 5a for all three spacecraft (for completeness, THD is also shown).

Here, we also observe a rather turbulent behavior at THD. For THA and THE, on the other hand, we see that $\Theta$ and $\varphi$ are relatively constant around $t_{\max}$ (marked with black circle). To obtain a direction for the jet velocity, we determined the mean value of $\Theta$ and $\varphi$ of the velocities measured by THA and THE around $t_{\max}$. This is shown in Figure 5b. For comparison, we have also plotted the direction of THD, but we can clearly see that it deviates strongly from THA and THE. As an error, we have calculated the standard deviation and also the maximum difference in $\Theta$ and $\varphi$. To be conservative, we use the maximum difference in $\Theta$ and $\varphi$ from the mean for error handling in the following analysis.

In the revised version we implemented these parts in lines 92-115 and Figures 2 and 3.

Later (in Figures 6 and 7) we show that the pattern of diverging and converging flows is not strongly dependent on the propagation direction, but is still visible when the errors are applied. In the revised version we will include the updated determination of the jet propagation direction together with the treatment of uncertainty.

In the revised version we implemented these parts in the Appendix A in lines 240-255 and Figures A1 and A2. The uncertainty treatment of the position of the central axis is done in Appendix B in lines 256-276 and Figure B1.

*Figure 2 is discussed only in terms of divergence or convergence of the flow prior to or after the jet passage but THE (that would be very close to the jet center - Figure 4) observes a relatively large velocity in the y direction. My understanding of the used coordinates is that it is the jet coordinate system and it is strange if its center moves with much larger velocity than its surrounding. The problem of perpendicular velocities is also clear in the discussion in lines 186-189. The authors claim the largest expansion at t_max but Figure 2 shows rather contraction in the perpendicular direction at this time.*

[Figure]

Figure 4: 1D distribution from integration along the other velocity axes.

We agree that we are only discussing the flow in qualitative terms. This was done because we cannot draw any conclusions from the absolute values of the velocity. The coordinate system used in Figure 2 (in the current manuscript) shows the two axes perpendicular to the assumed direction of jet propagation. In principle, this is intended to illustrate the flow anomaly caused by the jet (namely diverging and converging flows). The greater velocity at THE indicates a greater deviation from the background flow. The larger deviation near the center is in agreement with Plaschke and Hietala [2018]. The largest expansion of the jet that we show in Figure 4 (in the current manuscript) results from our fit to the dynamic pressure measurements. We claim that the intersection between the fitting function and the dynamic pressure threshold leads to an estimate of the size of the jet. Figure 2 (in the current manuscript) should only illustrate the plasma motion around the center, and the velocities do not necessarily correlate with the size of the jet. We will revise this section as we see the need to better describe our calculations and results so that readers can more easily understand our conclusions.

In the revised version we changed the description of Figure 2 (in the current manuscript). In the revised version this will be Figure 4 and the description is in lines 121-135.

*The analyzed interval in all figures is +/- 12 s from the jet center (dynamic pressure peak). The jet duration is nearly 60 s in observations of all spacecraft (Figure 1). The estimated perpendicular size is about 1 Re, the distances of the spacecraft from the estimated jet center are lower than 0.4 Re (Figure 3) and it means that the analysis of the velocity components in Figures 2 and 3 are done inside the jet. However, the authors claim that the analysis is based on flows around the jet induced by the jet propagation through the magnetosheath plasma but such flows should be analyzed close to but outside of the jet.*

The referee is absolutely right that the analyzed flow is inside the jet structure, and we thank the referee for pointing out the misleading description. Nevertheless, Plaschke and Hietala [2018] have shown that the evasive motion is also present inside the jet. Therefore, we should expect the same behavior with a diverging flow before and a converging flow after $t_{\mathrm{max}}$.

With the focus on the flow behavior within the jet, we saw the need to change our transformation to the jet coordinate system. The transformation involves only a simple rotation. We use the estimated propagation direction as the new $\mathbf{X}'$ axis and no longer subtract the background velocity. There are two reasons for this. Firstly, we only want to consider the velocity perpendicular to the propagation direction. Secondly, we do not see a bimaxwellian distribution (for THA and THE) and assume that the spacecraft observe almost exclusively the jet plasma.

We calculated $\mathbf{Y}'\left(=\frac{\mathbf{X}'\times\hat{\mathbf{X}}}{|\mathbf{X}'\times\hat{\mathbf{X}}|}\right)$ and $\mathbf{Z}'=\left(\frac{\mathbf{X}'\times\mathbf{Y}'}{|\mathbf{X}'\times\mathbf{Y}'|}\right)$ in the same way as described in the current manuscript, and we rotate the measured velocities and the positions of the spacecraft into the new coordinate system. The origin of our new coordinate system was chosen arbitrary to be at the position of the THA at $t_{\mathrm{max}}=[X=10.76R_{\mathrm{E}},Y=0.34R_{\mathrm{E}},Z=1.49R_{\mathrm{E}}]$ (in GSE coordinates). The results are shown in Figure 6 for the velocities determined from the peaks in the 1D VDFs and in Figure 7 for the velocities from the full moments for comparison. The middle rows in both figures show the resulting flow pattern when the mean jet propagation direction (see Figure 5b) is used. The bottom (top) rows of both figures show the resulting flow pattern when the errors are subtracted from (added to) the mean jet propagation direction.

[Figure]

(a) Time evolution of measured velocity direction over time for THA, THD and THE in the columns from left to right. The time is color coded with darker colors corresponding to earlier times and lighter colors corresponding to later times. The polar angle $\Theta$ and the azimuthal angle $\varphi$ are described in the text.

[Figure]

(b) Velocity direction around $t$max from THA, THD and THE in red, green and blue, respectively. The black dot corresponds to the mean value of the measurements from THA and THE.

Figure 5: Determination of propagation direction.

We continue to consider only THA (red arrows) and THE (blue arrows), as we have already mentioned that THD observes a mixture of plasma populations. We see in all cases in the time from -15s to -6s signs of diverging velocity vectors before $t_{\max}$. Around -3s to +9s, we observe high dynamic pressure (see Figure 1 in the current manuscript) and a more complex flow pattern. After $t_{\max}$ from +12s to +15s, we observe signs of the converging pattern, again in almost all cases. This should indicate the applicability of our approach even for a non-perfectly estimated propagation direction.

Furthermore, we estimated the position of the center of our jet from the diverging and converging flow at THA and THE. Figure 8 shows the temporal evolution of the center position for flow patterns determined from 1D VDFs and full moments. As we can see, the deviation of the center position is relatively small in both cases (except for errors added - top rows). In addition one can see that the center position is more stable when we use the velocities from the 1D-VDFs.

In the revised version of the manuscript we will alter the introduction and methodology accordingly and describe the new transformation together with the uncertainty treatment. We also want to point out that only two spacecraft are needed in principle to estimate the center of the jet.

In the revised version we implemented these parts in lines 115-151 and Figure 5.

[Figure]

Figure 6: Flow pattern inside the jet around $t_{\max}$ with velocities from 1D-VDFs. Dots (arrows) in red, green and blue represent the position (measured velocity) of THA, THD and THE, respectively. The middle row represent the resulting flow pattern when the mean jet propagation direction is used. The bottom (top) row shows the resulting flow pattern when the errors are subtracted from (added to) the mean jet propagation direction.

> *Also two-point results given in the conclusion are not surprising, everyone expects that from the definition of a jet.*

We agree that everyone expects this result. We still see the need to show it for individual events. In addition the shape of jets is still under debate and as shown in Figure 9 there are in principle multiple possibilities for the jet shape. In all columns fits for three different times (= different intersections of the jet) are shown. The darkest color correspond to a fit at $t_{\max}$ and lighter colors correspond to later times ($t_{\max} < t_1 < t_2$). Case a) belongs to a jet which has its largest expansion around $t_{\max}$. The expansion decreases towards the front and rear parts of the jet, while the dynamic pressure stays constant. In b) one can see the opposite case with constant expansion of the jet. The dynamic pressure is highest around $t_{\max}$ and decreases toward front and rear parts. In c) both dynamic pressure and expansion are highest around $t_{\max}$ and decrease before and afterwards. While we don't expect a) to be a realistic, we can't say for sure from a single spacecraft measurement which of the cases describes the jet shape. In addition, we also wanted to point out that statistical studies on magnetosheath jets systematically underestimate the dynamic pressure as spacecraft rarely observe the center of jets.

In the revised version, we point out that this is a case study showing how the method can help in determining jet sizes, and that the exact form here applies only to this jet. We still point out that there may be systematic biases in statistical studies. This occurs in lines 221-232.

> *Line 20 - I would suggest adding the sentence on the jet definition with corresponding reference here*

In the revised version we added this information in lines 25-27.

> *Line 23 - ........jet occurrence "on" solar wind parameters.......*

In the revised version we adapted this in line 30.

> *Line 30 - ...with the higher velocity and density..*

[Figure]

Figure 7: Flow pattern inside the jet around $t_{\max}$ with velocities from full moments. Dots (arrows) in red, green and blue represent the position (measured velocity) of THA, THD and THE, respectively. The middle rows represent the resulting flow pattern when the mean jet propagation direction is used. The bottom (top) row shows the resulting flow pattern when the errors are subtracted from (added to) the mean jet propagation direction.

We agree with the referee and will revise the manuscript accordingly.

*Line 39 - ...”jets can trigger and suppress reconnection”, this connection it seems contradictory, it needs to be explained*

We agree with the referee that we need more explanation. We intended to say that jets can modify the magnetic field by dragging the frozen-in field lines [Hietala et al., 2018]. With the modified magnetic field they can change the shear angle between magnetospheric and magnetosheath magnetic field lines [Hietala et al., 2018]. This could trigger reconnection under conditions where we would normally wont see it (e.g. northward IMF) and vice versa (e.g. southward IMF) [see also Vuorinen et al., 2021]. We will rewrite this point in the revised manuscript.

In the revised version we added a sentence in lines 45-47.

*Line 41 – I would suggest to skip .......”for a single event”......*

In the revised version we skipped this in line 49.

*Line 66 – ...”the central axis of this jet from evasive motion of the ambient plasma”, the sentence in this place is not clear*

In the revised version we updated the Introduction in lines 70-78.

*Line 70 and elsewhere in the text – “spin resolution” instead of spin fit resolution*

Done.

We agree with the referee and will revise the manuscript accordingly.

*Line 95 – in the description of Fig. 2, the link to Fig. 3 is not correct*

[Figure]

Figure 8: Time evolution of center position from the flow patterns with velocities from the 1D-VDFs (left column) and from the full moments (right column). The middle row represent the results when the mean jet propagation direction is used. The bottom (top) row shows the results when the errors are subtracted from (added to) the mean jet propagation direction.

We agree with the referee that we should not include information that were not already explained. We are restructuring sections 2 and 3 of the current manuscript to improve readability.

In the revised version we introduced the central axis to a later point in lines 142-151 and Figure 5. We don't reference this figure earlier.

**Author response to referee #2**

The authors thank Anonymous referee #2 for their comments on the manuscript. We will take the comments into account when revising the manuscript. Here we provide responses to each of the referee's comments (formatted as italics in indented paragraphs). We respond to the referee's comments not in the order in which they are listed in the original comment, but in an order that improves the readability of this response. At the end of our answers, we indicate in blue where the changes have been made in the revised version.

> Line 87: Vjet is defined between all SCs, but we can see from Figure 1, that velocity is different between the different THEMIS satellites. Also, what does "mean" value at t_max? isn't that 1 measurement, later it is suggested that it is the average value between the neighbor points, please clarify how the mean value is computed here.

$V_{\text{jet}}$ is calculated in the current manuscript as the mean value of one measurement of THA, THD and THE at their respective $t_{\max}$. We agree with the referee that THD observes a different velocity than THA and THE, as can be seen in Figure 1 in the current manuscript. The averaging over neighboring points was only done in Figure 2 to visualize the flow, not to estimate the propagation direction. In the revised manuscript, we will not average the velocities in order to make the paper easier to read. Moreover, the flow pattern is visible even without the averaging and we do not blur the measurements.

[Figure]

Figure 9: Hypothetical fits to dynamic pressure measurements plotted against distance from center for three different scenarios of jet shapes. In all columns fits for three different times (= different intersections of the jet) are shown. The darkest color correspond to a fit at $t_{\max}$ and lighter colors correspond to later times ($t_{\max} < t_1 < t_2$). The shown cases are described and explained in the text.

In the revised version we update Figure 2 (current manuscript). In the revised version this will be Figure 4. To calculate Vjet we use more measurements. This is described in lines 114-128 and Figure 3.

*Line 87: Furthermore, the velocity of the jet can be higher in some cases than the moment product, since jets and the background population may co-exist (See Raptis et al. 2022), making the velocity observations having lower values in the data. How would a larger Vjet change your coordinate system there if you either take different measurements (pairs of SCs, maximum values, etc.) ? Also, if you assume that the velocity of the jet population is actually higher than measured due to non-Maxwellianity features (so maybe + 30-40%)?*

We agree with the referee that only considering the full moments may be error prone. Therefore we have repeated our analysis considering the velocity distribution (VDF) and included error treatment. We started with a closer look at the velocity distribution functions. A first glance, the 2D slices show only single maxima for THA, THD and THE, as shown in Figure 1.

For a more detailed look, we used the similar approach as Raptis et al. [2022] and integrated the 3D distribution over two velocity axes to get a 1D velocity distribution along the third component. The 1D distributions along GSE-$V_{\mathrm{X}}$, -$V_{\mathrm{Y}}$ and -$V_{\mathrm{Z}}$ at the time of maximum dynamic pressure ($t_{\max}$) are shown in Figures 2, 3 and 4 for THA, THD and THE, respectively.

Here, we were not able to see a clear bi-Maxwellian distribution for THA and THE in either of the three components. Instead, we observe individual peaks, albeit with deviations from the ideal Maxwellian distribution. For THD, we clearly see two maxima in the 1D distribution along $V_{\mathrm{Y}}$ and $V_{\mathrm{Z}}$.
Following the same conclusion as Raptis et al. [2022], THA and THE observe one plasma population. THD, which is also further away from the other two, observes two plasma populations, probably a mixture of jet and magnetosheath background plasma. Together with the lower dynamic pressure at $t_{\max}$, this suggests that THD is closer to the edge of the jet. To be conservative, we continued our analysis using only THA and THE to determine the direction of jet propagation. Furthermore, since we see that the 1D VDFs at THA and THE deviate from ideal Maxwellians, we use the velocity at which the 1D VDF peaks in each component rather than the full plasma moments to determine the jet propagation direction. Other time steps inside the jet show similar 1D VDFs and outside the jet we observe 1D VDFs that look almost like ideal Maxwellians.

In the revised version we implemented these parts in lines 92-100 and Figure 2.

In the following (see figures 6 and 7) we will show that the coordinate system changes for different propagation directions, but we can still observe the diverging and converging flows.

In the revised version we implemented these parts in the Appendix A in lines 240-255 and Figures A1 and A2.

*Line 82-83: One direction vector is used, although it is very unlikely that it can fully character-ize the direction. How does the result change if you use direction vectors taken from only one of the spacecraft or different pairs of them?*

*Line 110: Here it is assumed that the propagation direction again is constant. This is not very likely to be the case as far as I am aware. How would this assumption change the result if it is not applicable? It would be nice to at least comment on that if quantification is not possible.*

We agree that one vector without uncertainty handling is not a good estimation of the propagation direction. We therefore investigated the stability of the direction of the jet velocity. Using the updated velocity values, we calculated the polar angle $\Theta$ ($= \arccos \frac{V_z}{\sqrt{V_x^2+V_y^2+V_z^2}}$) and the azimuthal angle $\varphi$ ($= \text{sgn}(V_y) \cdot \arccos \frac{V_x}{\sqrt{V_x^2+V_y^2}}$) of the measured velocities and plotted them over time. The results are shown in Figure 5a for all three spacecraft (for completeness, THD is also shown).

Here, we also observe a rather turbulent behavior at THD. For THA and THE, on the other hand, we see that $\Theta$ and $\varphi$ are constant around $t_{\max}$ (marked with black circle). To obtain a direction for the jet velocity, we determined the mean value of $\Theta$ and $\varphi$ of the velocities measured by THA and THE around $t_{\max}$.

This is shown in Figure 5b. For comparison, we have also plotted the direction of THD, but we can clearly see that it deviates strongly from THA and THE. As an error, we have calculated the standard deviation and also the maximum difference in $\Theta$ and $\varphi$. To be conservative, we use the maximum differ-ence in $\Theta$ and $\varphi$ from the mean for error handling in the following analysis.

In the revised version we implemented these parts in lines 101-115 and Figure 3. The uncertainty treatment of the position of the central axis is done in Appendix B in lines 256-276 and Figure B1.

*Line 94: Here the choice of 12 seconds before and after the tmax is not clear. Isn't that time within the jet observation itself since the duration of the jet is larger than that? Wouldn't that mean that for Figure 2 we just see the jet observations rather than the "before" and "after" periods of the jet flow? Something is unclear here.*

*Line 102-103: Again here, you are referring to an evasive motion, but shouldn't that occur "after" the jet observations? if that's only 12 seconds after tmax, aren't these observations still referring to the jet itself, and not the plasma after its passage?*

The referee is absolutely right that the analyzed flow is inside the jet structure, and we thank the referee for pointing out the misleading description. Nevertheless, Plaschke and Hietala [2018] have shown that the evasive motion is also present inside the jet. Therefore, we should expect the same behavior with a diverging flow before and a converging flow after $t_{\max}$.

With the focus on the flow behavior within the jet, we saw the need to change our transformation to the jet coordinate system. The transformation involves only a simple rotation. We use the estimated propagation direction as the new $\mathbf{X}'$ axis and no longer subtract the background velocity. There are two reasons for this. Firstly, we only want to consider the velocity perpendicular to the propagation direction. Secondly, we do not see a bimaxwellian distribution (for THA and THE) and assume that the spacecraft observe almost exclusively the jet plasma.

We calculated $\mathbf{Y}'$ $\left(= \frac{\mathbf{X}'\times\hat{\mathbf{X}}}{|\mathbf{X}'\times\hat{\mathbf{X}}|}\right)$ and $\mathbf{Z}' = \left(\frac{\mathbf{X}'\times\mathbf{Y}'}{|\mathbf{X}'\times\mathbf{Y}'|}\right)$ in the same way as described in the current manuscript, and we rotate the measured velocities and the positions of the spacecraft into the new coordinate system. The origin of our new coordinate system was chosen arbitrary to be at the position of the THA at $t_{\max} = [X = 10.76R_{\mathrm{E}}, Y = 0.34R_{\mathrm{E}}, Z = 1.49R_{\mathrm{E}}]$ (in GSE coordinates). The results are shown in Figure 6 for the velocities determined from the peaks in the 1D VDFs and in Figure 7 for the velocities from the full moments for comparison. The middle rows in both figures show the resulting flow pattern when the mean jet propagation direction (see Figure 5b) is used. The bottom (top) rows of both figures show the resulting flow pattern when the errors are subtracted from (added to) the mean jet propagation direction.

We continue to consider only THA (red arrows) and THE (blue arrows), as we have already mentioned that THD observes a mixture of plasma populations. We see in all cases in the time from -15s to -6s signs of diverging velocity vectors before $t_{\max}$. Around -3s to +9s, we observe high dynamic pressure (see Figure 1 in the current manuscript) and a more complex flow pattern. After $t_{\max}$ from +12s to +15s, we observe signs of the converging pattern, again in almost all cases. This should indicate the applicability of our approach even for a non-perfectly estimated propagation direction.

Furthermore, we estimated the position of the center of our jet from the diverging and converging flow at THA and THE. Figure 8 shows the temporal evolution of the center position for flow patterns determined from 1D VDFs and full moments.

As we can see, the deviation of the center position is relatively small in both cases (except for errors added - top rows). In addition one can see that the center position is more stable when we use the velocities from the 1D-VDFs.

In the revised version of the manuscript we will alter the introduction and methodology accordingly and describe the new transformation together with the uncertainty treatment. We also want to point out that only two spacecraft could suffice to estimate the center of the jet.

In the revised version we added the discussion of flow patterns in lines 125-138. The uncertainties are treated in Appendix A in lines 240-255 and Figures A1 and A2. The estimation of the central axis is seen in lines 139-151 and Figure 5.

*Line 86: Mean velocity is defined as the average velocity of a very large window, which is also not compared if it is similar between all SCs. Also, if there is variability due to the presence of other flows this should also be affected and we can actually see THD measuring a flow increase at 16:02, almost above the threshold given in dynamic pressure. How would your result change if you remove the jet observations and other potential fast flows or if you change the window you average through?*

The referee correctly points out that fast flows in the near jet environment have a strong influence on the calculated background flow. Using only quiet times (without increased dynamic pressure values), the background flow is reduced by 16% and rotates by 13°. However, since we will use a slightly different transformation, we no longer need the background flow.

In the revised version we present the updated calculation in lines 110-120 and Equation 2.

*Line 95: This is the first time the word "center" is used, dropped abruptly without much explanation as to what it means or how it is derived. Some re-structuring and clarification could be useful here.*

We agree that we have to rewrite the introduction and methodology according the updated analysis and with a structure that can be easier followed.

In the revised version we introduced the center at the end of the Data and Methods section with more explanation in lines 139-151.

*Readability of the paper: My overall suggestion here is to restructure section 2 and 3 to form clearer sections. Presently, neither of these sections are clearly separated, which forces the reader to go back and forth between the sections to understand what you are trying to say. For example, Figure 2 is only mentioned in the data and methods section, but then it is presented as part of the results in the discussion. The Gaussian fit is suddenly mentioned in the result section, which should have been mentioned before. Also, the motivation for this fit profile should have been made in the data and methods section, but currently it is shown at the end of the discussion (lines 168-173)! A re-organizing of such aspects could greatly improve the readability of the paper.*

We agree that the structure of the paper needs to be revised. In the revised version of the manuscript, we will focus on a more consistent structure that should be easier for the reader to follow.

In the revised version we separated clearer between Data and Methods and the Result section. We introduced the Gaussian Distribution earlier in lines 152-160 and Equation 3.

*Figure 2: You mention that the top axis shows the Y' coordinate, while the Z' is constant. However, the values shown on the top x-axis appear to be constant on every time step. Could you explain a bit more what we are seeing here?*

The $Y'$ axis is also the same for each time step. In contrast to the $Z'$ axis, we have plotted the $Y'$ axis separately for each time step, so that the reader can in principle see the actual positions of the spacecraft in the flow chart. We recognize the need to describe the figure in a better way and will do so in the revised manuscript.

In the revised version we updated the description of Figure 2 (in the current manuscript). In the revised version this will be Figure 4. The changes are in lines 121-135.

*Lines 177-180: These two results, appear to be driven by the definition of a jet and the methodology itself. In other words, Tmax will automatically drive a dynamic pressure higher, since, well, the fit parameters at t_max have as input higher dynamic pressure pairs, so Po will be higher. Is that correct? Do I miss something? Also, the same applies for $\Delta R$. According to my understanding, this will be the cause for every jet that has a single peak in dynamic pressure (and therefore lower values before and after it). Could you either help me understand if I miss something here, or describe a hypothetical (or real) case for which your methodology would not result always in the same conclusion*

From the measurement of a single spacecraft one can't say anything about the jet shape as the region and direction how the spacecraft traverses the jet has a great influence on the measurement. With our methodology we try to gather more information about the shape of the jet from two or more spacecraft measurements. The shape of jets is still under debate and as shown in Figure 9 there are in principle multiple possibilities for the jet shape. In the Figure, in all columns the darkest color correspond to a fit at $t_{\max}$ and lighter colors correspond to later times ($t_{\max} < t_1 < t_2$). Case a) belongs to a jet which has its largest expansion around $t_{\max}$. The expansion decreases towards the front and rear parts of the jet, while the dynamic pressure stays constant. In b) one can see the opposite case with constant expansion of the jet. The dynamic pressure is highest around $t_{\max}$ and decreases toward front and rear parts. In c) both dynamic pressure and expansion are highest around $t_{\max}$ and decrease before and afterwards. While we don't expect a) to be a realistic, we are can't say for sure from a single spacecraft measurement which of the cases describes the jet shape. In addition, we also wanted to point out that statistical studies on magnetosheath jets systematically underestimate the dynamic pressure as spacecraft rarely observe the center of jets. We realize that we need to rewrite the last section to more clearly present our conclusion to the reader.

In the revised version we changed the Summary and Conclusion section in lines 216-238.

*Line 2: Not all jets are formed at the bow shock, maybe such a general statement shouldn't be in the abstract.*
*Line 3-6: This sentence is very hard to read, consider splitting it.*

In the revised version we updated the Abstract in lines 1-13.

*Line 13 and throughout the text: Consider changing $\Theta$ to $\Theta\_Bn$ since that's the typical symbol used.*

Done in lines 18-19.

We agree with the referee and revise the manuscript accordingly.

**List of relevant changes**

- Changed title

- Changed summary

- Last paragraph of the introduction changed

- Two new figures and their description added in the section Data and Methods to address the uncertainties of the measurements

- Equation for polar coordinates for estimating the propagation direction $V_{\text{jet}}$ added

- Changed equation for calculating the coordinate system

- Changed description of the flow pattern

- Changed mapping for estimating the centerline and the description

- Equation for the fit has been moved to the Data and Methods section

- Results and Discussion merged into one section

- Figure with dynamic pressure profiles modified and treatment of uncertainty added in Results and Discussion section

- Stronger emphasis on demonstration of the method in the Summary and Conclusion section

- Appendix A and Appendix B added for treatment of uncertainties

**References**

Hietala, H., Phan, T. D., Angelopoulos, V., Oieroset, M., Archer, M. O., Karlsson, T., and Plaschke, F. (2018). In situ observations of a magnetosheath high-speed jet triggering magnetopause reconnection. *Geophys. Res. Lett.*, 45(4):1732–1740. doi: 10.1002/2017GL076525.

Plaschke, F. and Hietala, H. (2018). Plasma flow patterns in and around magnetosheath jets. *Ann. Geophys.*, 36(3):695–703. doi: 10.5194/angeo-36-695-2018.

Raptis, S., Karlsson, T., Vaivads, A., Lindberg, M., Johlander, A., and Trollvik, H. (2022). On magnetosheath jet kinetic structure and plasma properties. *Geophysical Research Letters*, 49(21):e2022GL100678. doi: 10.1029/2022GL100678.

Vuorinen, L., Hietala, H., Plaschke, F., and LaMoury, A. T. (2021). Magnetic field in magnetosheath jets: A statistical study of bz near the magnetopause. *Journal of Geophysical Research: Space Physics*, 126(9):e2021JA029188. e2021JA029188 2021JA029188.

---

## Author Response (AR2)

In this document we give responses to comments from both referees. We start with referee #1 and then answer referee #2.

**Author response to referee #1**

The authors thank Anonymous referee #1 for their comments on the manuscript. We will take the comments into account when revising the manuscript. In this document we provide responses to each of the referee's comments (formatted as italics in indented paragraphs). At the end of our answers, we indicate in blue where the changes have been made in the revised version.

> *1) The authors added the analysis of VDF bringing interesting results but the discussion of them is insufficient. There are several features that should be clarified because they can influence the interpretation of results:*
>
> > - *The authors conclude that only distribution in the panel b2 exhibits two peaks. However, even a very brief visual analysis reveals two populations in a1, a2 and c1 panels.*
> >
> > - *The distribution of Vx in a1 and c1 peaks at the jet velocity, whereas the distribution in b1 peaks at the background magnetosheath velocity. It implies that THD is outside the jet proper even at the maximum of the dynamic pressure. A possible explanation could be the compression of the ambient magnetosheath plasma that leads to increase of the dynamic pressure.*
> >
> > - *The rest of the ion population with a low (magnetosheath) velocity can be identified also in a1 and c1 panels.*
> >
> > - *Distribution of Vz has two peaks in b2 but similar two populations are visible in a2.*
> >
> > - *The last three points suggest that the plasma inside the jet is a mixture of original magnetosheath population with the incoming jet population.*

We agree with the referee that the figure and its implications should be discussed in more detail. We also can not rule out that THD is observing the ambient magnetosheath rather than the jet. In the following responses, we will go into more detail about the jet velocity determination and its impact on the results. In the revised version the discussion of Fig. 2 is updated in lines 100-111.

> *2) Since VDF inside the jet is non-Maxwellian and consists of two populations with very different velocities, the velocity determined as the first-order moment strongly depends on their relative abundance. This abundance would vary in space and time and it can affect the analysis of velocity direction that is discussed in Figures 3 and 4 and used in determination of the jet dimensions.*

We agree with the referee that the VDFs may vary temporally and spatially. Therefore, we manually inspected the 1D VDFs at each time step within the jets and selected the jet velocity in the following way: If we observe two populations in the 1D VDFs (two peaks or a strong deviation from the ideal Maxwellian curve), we choose the highest absolute velocity in the x-direction and for the y- and z-directions the coldest population (peak with steeper slope and smaller width) [similar to Raptis et al., 2022].

When comparing the manually selected velocities with our previous approach of using the velocity at which the 1D VDFs peak, we find almost no difference for THA and THE. This means that we have already successfully selected the velocity of the jet plasma population. For THD, we observe more deviations. However, this has no influence on the analysis of the flow pattern as we only use THA and THE measurements. However, it should have an impact on the dynamic pressure profile as the measured dynamic pressure is affected at THD, as the referee notes in the next comment. In the revised version we describe the velocity determination in lines 112-118. We updated Fig. 3, 4, 5 and 6 and the figures in the appendices as velocities at THD have changed marginally.

*3) The size analysis in Figure 6 would take into account the fact that the dynamic pressure enhancement observed by THD is caused by an increase of the density whereas the velocity increase is the major contributor to the dynamic pressure observed by other two spacecraft.*

We agree with the referee that the density increase at THD can also be explained by the ambient magnetosheath plasma being compressed by the jet and thus implying that THD is not observing the jet. Therefore, we fitted the Gaussian profile to both the THA and THE measurements only and to the measurements for all three spacecraft with updated THD measurements (manually selected as described above). The qualitative behavior and our conclusion remain the same. This is shown as an example for three profiles in Fig. 1 for the fit to all three spacecraft and in Fig. 2 for the fit to THA and THE only (note that we chose a different time step than in the current manuscript after $t_{\mathrm{max}}$ as the Gaussian profile was not applicable for this case).

[Figure]

Figure 1: Dynamic pressure $P_{\mathrm{dyn,x}}$ derived from velocities from the 1D VDFs in the spacecraft system versus the distance from the center $r$ at THA, THD and THE (crosses in red, orange and blue, respectively) at 9 s before $t_{\mathrm{max}}$ (a), at $t_{\mathrm{max}}$ (b) and at 15 s after $t_{\mathrm{max}}$ (c). The black, dashed line represent a fit with a Gaussian distribution to the data points of all three spacecraft and the gray area visualizes when we subtract/add one standard deviation $\sigma$ from the optimal fit parameters. The blue horizontal line depicts a quarter of the solar wind dynamic pressure.

[Figure]

Figure 2: Dynamic pressure $P_{\mathrm{dyn,x}}$ derived from velocities from the 1D VDFs in the spacecraft system versus the distance from the center $r$ at THA, THD and THE (crosses in red, orange and blue, respectively) at 9 s before $t_{\mathrm{max}}$ (a), at $t_{\mathrm{max}}$ (b) and at 15 s after $t_{\mathrm{max}}$ (c). The black, dashed line represent a fit with a Gaussian distribution to the data points of THA and THE. The blue horizontal line depicts a quarter of the solar wind dynamic pressure.

We can see that the Gaussian profiles 9s before and 15s after $t_{\mathrm{max}}$ are quite similar. The profile at $t_{\mathrm{max}}$ differs more in both figures, but the conclusion of higher perpendicular size and higher dynamic pressure is visible in both. In the revised manuscript, we will continue to fit the Gaussian profile to all three spacecraft measurements because the qualitative results are the same and we can provide estimates for the uncertainty of the fit, which is not possible with only two data points. However, we will discuss the possiblity of THD not observing the jet but the ambient magnetosheath. We will update Figure 6 and Figure B1 in Appendix B in the revised version. We will also explain the manual verification by selecting the velocity of the jet population as described above.

In the revised version we add the parts in lines 108-110 and 218-224.

> *The abstract would contain major results, it is not a proper space for discussion or description of the methods.*

We agree and will change the abstract accordingly.
In the revised version the updated abstract is in lines 1-11.

> *Line 43 – maybe it would probably be appropriate to also mention the paper by Nemecek et al. (2023) which illustrates the importance of jets for magnetopause processes.*

We agree and will add this part to the revised version.
In the revised version this is done in lines 42-44.

> *Figure 1 – there are no dashed vertical lines*

We will enhance the line width and separation to make the dashed lines better visible.
In the revised version Figure 1 is updated.

> *Line 39 - . . . "jets can trigger and suppress reconnection", this connection it seems contradictory, it needs to be explained*

We believe this is a leftover from the first referee report as in the actual version of the manuscript this sentence is in line 45-46 and has the following explanation: "Hietala et al. [2018] showed that jets can trigger and suppress reconnection at the magnetopause, as they can modify the magnetic field in the magnetosheath and thus alter the shear angle at the magnetopause."
If the referee is not satisfied with the explanation, please let us know.
In the revised version this can be found in lines 44-46.

> *Line 183 – the brackets are probably at a bad location*

We agree with the referee and will split and rewrite this sentence to make it easier to read.
In the revised version this is done in lines 212-214.

> *Lines 101 and farther – It is not clear which velocities were used in the calculations – standard onboard moments or velocities determined by authors using analysis of 1D distributions. If the second is true, the method of determination of velocity components would be briefly discussed.*

In lines 99-100, we explain that we will use the velocity at which the 1D VDFs peak for the analysis. We will give a more detailed explanation in the revised manuscript and alter/remove the reference to Fig. 1 where we show moment data to avoid confusing the reader.
In the revised version the determination of the velocities from the 1D VDFs is discussed in lines 112-118.
In lines 119-121 we are now referring to velocities from 1D VDFs.

**Author response to referee #2**

The authors thank Anonymous referee #2 for their comments on the manuscript. We will take the comments into account when revising the manuscript. In this document we provide responses to each of the referee's comments (formatted as italics in indented paragraphs). At the end of our answers, we indicate in blue where the changes have been made in the revised version.

> *1. The dynamic pressure in the central part of the jet is higher at tmax (6 nPa) and decreases towards the front and rear. So, from the 3 cases presented in your reply (Figure 9) (b) and (c) have a decreased dynamic pressure towards the front and rear. The case (a), as also stated in the reply, is unrealistic, and I fully agree. We already know that it is unrealistic from previous studies.*
> *Therefore, this conclusion is bounded to happen and is driven by the methodology as written in*

*158-159 and 210-21. Is that correct?*

The methodology only leads to a monotonic decrease of $P_{dyn}$ with greater distance from the central axis $r$ in the direction perpendicular to the jet propagation (since this monotonic decrease is inherent to the Gaussian fit). The lower dynamic pressure at the front and rear (along the propagation direction) are actual results of the measurements and the only realistic explanation if the time series exhibit a peak in the center. If the time series exhibit additional peaks in the front and rear part, the central dynamic pressure would increase again, as we will show in the revised version for our event.
In the revised version we explained our assumptions in more detail in lines 176-181.

> *Similarly, lines 176-177, seems to always be true for single-peak dynamic pressure jets? Or if you pick times that a monotonic relationship applies? (i.e., skipping the second peak in dynamic pressure shown in 16:04:30 on THE?*
> *If that's the case, maybe rather than stating a tautology, it can be more interesting to comment on some more details? For example, the decay of dynamic pressure before, after and at tmax seems to drop under slightly different rate with respect to distance. Could this be due to evolution in these small-time scales for the jet? Could it be something else? One could comment on the slope of these curves and have some discussion or ideas for future work.*
> *If I misunderstood again your argument, it would be very useful to include some hypothetical timeseries of a jet that you would have to be observed by a spacecraft that would give you a different conclusion than the one you present.*

We assume a monotonic decrease from the central axis towards the edge (in the direction perpendicular to the jet propagation). We make no assumptions for the dynamic pressure along the central axis (in the direction parallel to the jet propagation). We can also apply the method to multi-peak jets, and the referee rightly points out that there is a second peak after $t_{max}$ (12 s afterwards). This peak is visible, for example, in Appendix Figure B1 (a1). Here you can see that $P_0$ rises 12 s after $t_{max}$. Thus, even more peaks in the dynamic pressure measurements lead to a more complex 3D shape of the jet.

We will add the second peak to the discussion. On the other hand, the referee gives a good suggestion for potential future works.
In the revised version we updated Fig.6 and included the time line of $P_0$ and $\Delta R$. We discuss the shape in more detail in lines 193-224. We also discussed potential future works in lines 267-270.

> *It should be noted that the jet discussed is a rare event. With relatively low velocities and very high densities at the order of 100 [1/cc]. Especially, these high values on the number density are describing a not so average jet. This happens due to the upstream density being very high during that interval. While I don't see any immediate effect on the results, it should potentially be mentioned and discussed very briefly in the manuscript. Jets can have different conditions under different solar wind drivers, and therefore potentially different shapes.*

We agree that this is a rare event. In lines 233-234 of the current manuscript, we have already written that this jet event belongs to only a fraction of the jets that show clear signs of the flow pattern. The density of the solar wind upstream is at the upper end of the distribution with 11-12 [1/cc] [see Fig.1 Ma et al., 2020]. Looking at the density in the magnetosheath, we find that the density in the ambient plasma is about four times the upstream density and an 8-12 times higher density is observed in the jet (increase of 100-200% from the magnetosheath to the jet). This is relatively high, but should not affect our conclusion as we used the Plaschke criterion, which uses the solar wind dynamic pressure as a threshold. The velocities are not so low given the fact that we are closer to the magnetopause ($X=11R_E$) and still observe velocities above half the solar wind values (350 km/s) around the jet core.

Other events may differ quantitatively, but we see no reason why it should not work for them. Therefore, the method should also be applicable to "weaker" jets.

In the revised version, we will discuss the jet and its properties in more detail and emphasize that the shape we obtain for this case should not be taken as a general shape. We will put even more emphasis on the methodology than on the explicit result for this jet.
In the revised version we added parts to the description of the jet in lines 92-99 and expanded the discussion in the Summary and Conclusion section in lines 278-282.

> *I am glad that the authors clarified the "evasive maneuver" part of their work and on previous*

*studies. It appears, both myself and Referee #1 were confused by the terminology. The jet appears to have a variable velocity, but also variable magnetic field. Could this evasive maneuver primarily the jet being field aligned and following the magnetic field variation? At least THA/THE shows a correlation between the components of B and v.*

We agree that there is some correlation between the components of B and V. Compared to the ambient magnetosheath, the correlation within the jet structure is higher. However, the correlation coefficients for the individual components within the jet interval are not high for THA/THE except for the GSE-Z component (THA: $x = 0.50, y = 0.56, z = 0.80$ and THE: $x = 0.65, y = -0.34, z = 0.92$). In comparison, the correlation coefficients in the ambient plasma are between $-0.13$ and $0.55$.

In our opinion, this is in agreement with Plaschke et al. [2020, 2017] and their description of a tendency for the alignment of B and V in jets. They argue that the reason for this may be the high speed at which jets move through the slower ambient magnetosheath plasma, thereby straightening the magnetic field in their path. And also here we would argue that the plasma flow affects the magnetic field as the plasma beta $\beta(= p_{\mathrm{therm}}/p_{\mathrm{mag}})$ is always above 1 within in the jet interval and even the ambient magnetosheath (median values of $\beta$ in jet interval for THA and THE are 8.22 and 11.84, respectively; median values in ambient magnetosheath for THA and THE are 15.96 and 37.14, respectively). The mean values are even higher. This hints that the magnetic field is not strong enough to significantly alter the plasma flow.

The discussion about the possible correlation of $V$ and $B$ is treated in lines 95-99.

*Lines 231-232: That's something relatively obvious, and the community is (hopefully) aware. Always good to remind, though. Maybe this is also a good point to highlight that to compare observations with simulation this needs to be considered for all mesoscale transients and localized phenomena in every plasma environment.*

That is a great comment and we will add it in the revised version.

In the revised version this is done in lines 271-276.

*Line 110: "variate not too much" is a bit weird, perhaps rephrase to "do not variate significantly" or something along these lines.*

We agree and will change the sentence in the revised version.

In the revised version this is done in line 128.

*Lines 237-238: It would be nice to be consistent and reference the mission articles for all the missions mentioned there.*

We agree and will add the corresponding references.

In the revised version this is done in lines 285-287.

**List of relevant changes**

- Changed Abstract

- Added more discussion to Figure 2

- Changed the velocity determination according to Raptis et al. [2022]

- Changed Figures 3,4 and 5 due to new velocities

- Added 2 panels and more discussion to Figure 6

- Added discussion in Summary and Conclusion

- Changed Figures in Appendix A and Appendix B due to new velocities

**References**

Hietala, H., Phan, T. D., Angelopoulos, V., Oieroset, M., Archer, M. O., Karlsson, T., and Plaschke, F. (2018). In situ observations of a magnetosheath high-speed jet triggering magnetopause reconnection. *Geophys. Res. Lett.*, 45(4):1732–1740. doi: 10.1002/2017GL076525.

Ma, X., Nykyri, K., Dimmock, A., and Chu, C. (2020). Statistical study of solar wind, magnetosheath, and magnetotail plasma and field properties: 12+ years of themis observations and mhd simulations. *Journal of Geophysical Research: Space Physics*, 125(10):e2020JA028209.

Plaschke, F., Jernej, M., Hietala, H., and Vuorinen, L. (2020). On the alignment of velocity and magnetic fields within magnetosheath jets. *Ann. Geophys.*, 38(2):287–296. doi: 10.5194/angeo-38-287-2020.

Plaschke, F., Karlsson, T., Hietala, H., Archer, M., Vörös, Z., Nakamura, R., Magnes, W., Baumjohann, W., Torbert, R. B., Russell, C. T., and Giles, B. L. (2017). Magnetosheath high-speed jets: Internal structure and interaction with ambient plasma. *J. Geophys. Res.-Space*, 122(10):10,157–10,175.

Raptis, S., Karlsson, T., Vaivads, A., Lindberg, M., Johlander, A., and Trollvik, H. (2022). On magnetosheath jet kinetic structure and plasma properties. *Geophysical Research Letters*, 49(21):e2022GL100678. doi: 10.1029/2022GL100678.

---

## Author Response (AR3)

**Author response to referee #2**

The authors thank Anonymous referee #2 for their comments on the manuscript. We will take the comments into account when revising the manuscript. In this document we provide responses to each of the referee's comments (formatted as italics in indented paragraphs). At the end of our answer, we indicate in blue where the changes have been made in the revised version.

> *Fig. 6e – How the mean $\Delta R$ was determined? It seems to be two orders of magnitude larger at times +6 s and +9 s than at other times. If the jet plasma expands to such a large cross-section, it would be strongly rarefied and thus it cannot posse the jet patterns. However, panel 6d shows a large pressure. Where the plasma filling this large volume comes from?*

$\Delta R$ and similarly $P_0$ were determined by fitting the Gaussian distribution (Eq. 3 in the manuscript) to the data points at each time step. The Gaussian fit was not appropriate for the two time points +6 s and +9 s because THD observed a higher dynamic pressure than THA, even though THD was farther away from the central axis. Therefore, these values have large errors (indicated by the large gray areas) and should not be considered as a large expansion of the jet perpendicular to the propagation direction. We see the need to describe and explain Figure 6 in more detail.

> *Line 222 – Fig. 6d in this line is probably a misprint. The perpendicular jet extent is in Fig. 6e.*
> *Line 223 – "But the shape is still quite complex as we also observe contrary increases and decreases of P0 and $\Delta R$." What this sentence means? Does it refers to the apparent jet expansion in times +6 and +9 s?*

We wanted to point out that the perpendicular size (intersection of the fit with the threshold) depends on the dynamic pressure of the central axis $P_0$ and the width of the Gaussian fit $\Delta R$. We admit that the discussion is not described clearly enough and might confuse the reader. We see the need to describe our discussions and conclusions on Figure 6 in more detail.

> *Line 224 – In my understanding, tmax -9 s is probably a misprint, a large perpendicular extension is observed at tmax +9 s.*

This is not a misprint. We wanted to point out that the perpendicular size (intersection of the fit with the threshold) in Fig. 6a is larger than in 6c, although $P_0$ in Fig. 6c is larger than in 6a.

The description and discussions of Figure 6 were expanded and rewritten in lines 198-222 in the revised manuscript.